# Natural statistics support a rational account of confidence biases

Taylor W. Webb ®[1] ✉, Kiyofumi Miyoshi ®[2], Tsz Yan So[3], Sivananda Rajananda[4] & Hakwan Lau ®[5] ✉

Previous work has sought to understand decision confidence as a prediction of the probability that a decision will be correct, leading to debate over whether these predictions are optimal, and whether they rely on the same decision variable as decisions themselves. This work has generally relied on idealized, low-dimensional models, necessitating strong assumptions about the representations over which confidence is computed. To address this, we used deep neural networks to develop a model of decision confidence that operates directly over high-dimensional, naturalistic stimuli. The model accounts for a number of puzzling dissociations between decisions and confidence, reveals a rational explanation of these dissociations in terms of optimization for the statistics of sensory inputs, and makes the surprising prediction that, despite these dissociations, decisions and confidence depend on a common decision variable.

When faced with a decision, we have the ability not only to choose from a set of possible options, but also to assess how confident we are in our choice. This capacity is an important part of the decision-making process, allowing us to decide whether to gather more information[1], or how much to wager on the outcome of a decision[2]. It has been proposed that this sense of confidence corresponds to an optimal prediction of the probability that a decision will be correct, and that confidence is computed based on the same underlying decision variable as decisions themselves[3–9]. Given certain distributional assumptions, this approach entails the use of a decision variable that is proportional to the balance-of-evidence (BE; Fig. 1a), incorporating sensory evidence both for and against a decision[10]. This optimal view of confidence, however, has been called into question by a number of puzzling dissociations between decisions and confidence. These dissociations have led to the formulation of an alternative model in which decisions are made according to the BE rule, but confidence is estimated using a simpler heuristic strategy that primarily considers the response-congruent-evidence (RCE; Fig. 1b)[11–17]. That is, after weighing the evidence and making a decision, confidence is based mainly on the evidence in favor of the decision that was made.

These findings raise the question of why confidence would be computed using an apparently suboptimal heuristic. This is especially puzzling given findings suggesting that decisions are based on the balance of evidence[15], because it suggests that the evidence against one's choice is available in the decision-making process, but simply not incorporated into confidence judgments. One potential avenue for resolving this puzzle is to reconsider the assumptions underlying the low-dimensional models that have been employed in previous work. In particular, it has been shown that, given alternative assumptions about the variance structure governing stimulus distributions, the optimal approach to estimating confidence entails a more complex function that differs from both the BE and RCE rules (Fig. 1c)[18], with some evidence that human decision confidence follows this pattern[19]. However, it has yet to be shown whether this alternative model can account for the previously observed dissociations between decisions and confidence. More importantly, this alternative model calls attention to the fact that questions about optimality must be framed in relation to stimulus distribution structure, which has typically been treated as a modeling assumption in previous work.

In this work, we developed a model of decision confidence that operates directly on naturalistic, high-dimensional inputs, avoiding the

[1]University of California, Los Angeles, CA, USA. [2]Kyoto University, Kyoto, Japan. [3]The University of Hong Kong, Hong Kong, Hong Kong. [4]Harvard University, Cambridge, MA, USA. [5]Laboratory for Consciousness, RIKEN Center for Brain Science, Saitama, Japan. ✉e-mail: taylor.w.webb@gmail.com; hakwan@gmail.com

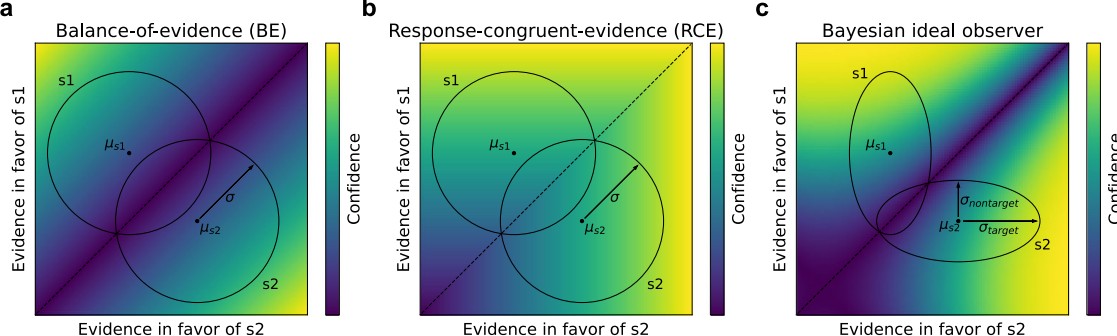

**Fig. 1 | Detection-theoretic formalization of confidence in two-choice tasks.** **a** Stimuli are modeled as samples from two-dimensional Gaussian distributions (with means $\mu_{s1}$ and $\mu_{s2}$, and variance $\sigma$), schematized as circles labeled s1 and s2, where each dimension represents the evidence in favor of one stimulus category. Given these assumptions, the optimal procedure for estimating confidence is a balance-of-evidence (BE) rule, based on the difference between the evidence in favor of s1 and s2. **b** Many results are well modeled by an alternative response-congruent-evidence (RCE) heuristic, according to which, after making a decision, confidence is based entirely on the evidence in favor of the chosen stimulus category, ignoring the evidence in favor of the alternative choice. **c** Bayesian ideal observer with alternative variance assumptions. When stimulus distributions are characterized by greater variance in the dimension in favor of the correct answer ($\sigma_{target}$) than the dimension in favor of the incorrect answer ($\sigma_{nontarget}$), as proposed in refs. 18 and 19, the optimal procedure for estimating confidence involves a more complex function.

need for these simplifying assumptions. To do so, we first developed a performance-optimized neural network model trained both to make decisions from high-dimensional inputs, and to estimate confidence by predicting the probability those decisions will be correct. Surprisingly, a number of seemingly suboptimal features of confidence naturally emerged from the model, including the positive evidence bias. We then used unsupervised deep learning methods to extract a low-dimensional representation of the model's training data. We found that the training data distribution displayed key properties that undermined the presumed optimality of the BE model, and that an ideal observer applied to this distribution replicated the observed dissociations, thus yielding a rational account of these dissociations. Consistent with this, we found that altering the distribution of the training data altered the resulting biases in predictable ways, and that the model employed a common internal decision variable for both decisions and confidence, despite the observed behavioral dissociations. Finally, we found that the model also accounts for a range of neural dissociations between decisions and confidence, including some features akin to blindsight resulting from lesions to the primary visual cortex.

## Results

Figure 2 illustrates the architecture and training data for our performance-optimized neural network model of decision confidence. The model was trained through standard supervised learning methods both to make a decision about (i.e., classify) an input image, and also to predict the probability that its own decision was correct (Fig. 2a). The model was trained on two standard image classification benchmarks, the MNIST and CIFAR-10 datasets, using supervised learning, and was also trained on an orientation discrimination task using reinforcement learning (RL) (Fig. 2b). Both contrast and noise level were varied during training, to give the model exposure to a broad range of conditions (Fig. 2c). See "Methods" for more details on the model and training procedures.

### Behavioral dissociations between decisions and confidence

We first assessed whether the model accounted for a number of previously reported behavioral dissociations between decisions and confidence. Despite not being directly optimized to produce these dissociations, the model naturally accounted for them in a manner that closely resembled the previous findings.

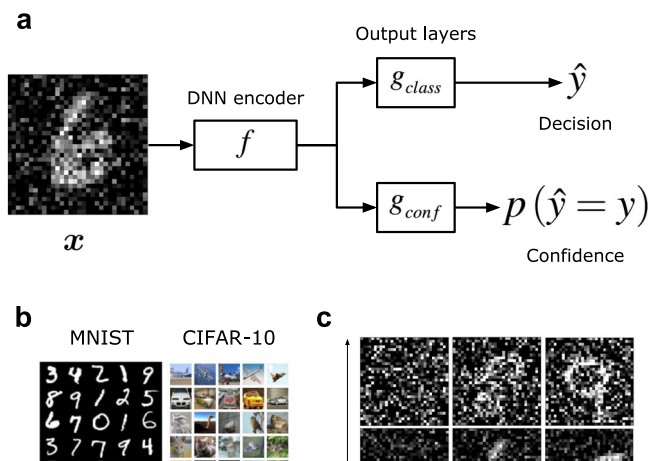

**Fig. 2 | Performance-optimized neural network model of decision confidence.** **a** Model architecture. An image **x**, belonging to class $y$, was passed through a deep neural network (DNN) encoder $f$ (the exact architecture of this encoder depended on the dataset, as detailed in "Encoder"), followed by two output layers: $g_{class}$ generated a decision $\hat{y}$ classifying the image, and $g_{conf}$ generated a confidence score by predicting $p(\hat{y} = y)$, the probability that the decision was correct. **b** The model was trained through supervised learning using the MNIST handwritten digits dataset and the CIFAR-10 object classification dataset. For these datasets, the classification layer was trained to label the class of the image, and the confidence layer was trained to output a target of 1 if the classification response was correct, and 0 if the classification response was incorrect. The model was also trained through reinforcement learning (RL) to perform an orientation discrimination task, in which, rather than generating an explicit confidence rating, the model had the option to opt-out of a decision and receive a small but guaranteed reward, allowing the use of the opt-out rate as an implicit measure of confidence. **c** To evaluate the relative influence of signal strength and noise on the model's behavior, images were modified by manipulating both the contrast $\mu$ and the noise level $\sigma$.

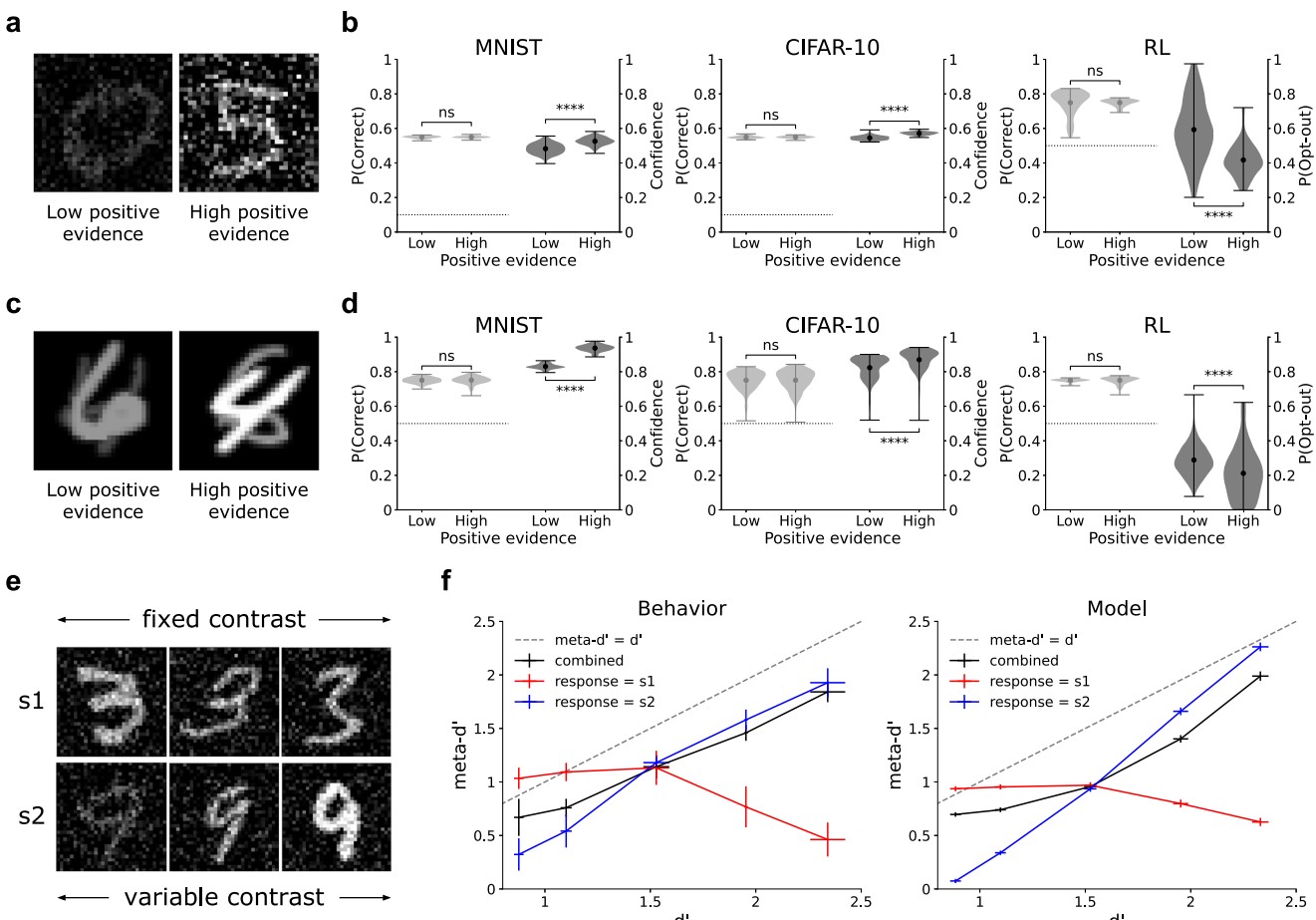

**Fig. 3 | Behavioral dissociations between decisions and confidence. a** Human and animal decision confidence displays a positive evidence (PE) bias: higher confidence (or lower opt-out rate) in the high vs. low PE conditions despite balanced signal-to-noise ratio and balanced decision accuracy. **b** The PE bias naturally emerges in performance-optimized neural networks across multiple datasets, architectures, and learning paradigms (two-sided paired t-tests; accuracy: MNIST, $p = 0.42$; CIFAR-10, $p = 0.48$; RL, $p = 0.97$; confidence: MNIST, $p = 1.4 \times 10^{-20}$; CIFAR-10, $p = 4.3 \times 10^{-50}$; RL, $p = 4.4 \times 10^{-20}$). See Supplementary Fig. S1 for confidence distributions in correct vs. incorrect trials. Note that stimulus parameters (contrast and noise) were set so as to target the threshold between chance performance (dotted black lines) and 100% accuracy, resulting in ~55% accuracy for 10-choice tasks and ~75% accuracy for two-choice tasks. The model achieved much higher accuracy when presented with noiseless images (96.3% ± 0.03 for MNIST, 88.1% ± 0.05 for CIFAR-10). **c** An alternative test for the PE bias, involving superimposed stimuli presented at different contrast levels, where the task is to indicate which stimulus is presented at a higher contrast. In the high positive evidence condition, there is both higher positive evidence (evidence in favor of the correct

answer, 4 in this case), and higher negative evidence (evidence in favor of the incorrect answer, 6 in this case), than in the low positive evidence condition. Visual noise was also included in images, but is omitted here for clarity of visualization. **d** The model also shows this alternative formulation of the PE bias (two-sided paired t-tests; accuracy: MNIST, $p = 0.8$; CIFAR-10, $p = 0.99$; RL, $p = 0.83$; confidence: MNIST, $p = 8.3 \times 10^{-84}$; CIFAR-10, $p = 2.8 \times 10^{-48}$; RL, $p = 2.8 \times 10^{-10}$). **e** Adaptation of behavioral paradigm from Maniscalco et al.[13], s1 is presented at an intermediate contrast, while the contrast of s2 is varied. **f** This produces a strong dissociation between type-1 sensitivity (d') and type-2 sensitivity (meta-d'): when participants respond s1, meta-d' decreases as d' increases (Behavior), a phenomenon which is captured by the neural network model (Model). Results in (**b**) and (**d**) reflect probability density over 100 trained networks, with mean accuracy/confidence in each condition represented by circular markers, and maxima/minima represented by the upper/lower line; results in (**f**) reflect mean d'/meta-d' over 100 trained networks ± the standard of the mean; ns indicates $p > 0.05$, ****$p < 0.0001$. Source data are provided as a Source Data file.

**The positive evidence bias.** Confidence is characterized by a positive evidence (PE) bias[11,12,14,16,17], as revealed by two related, but distinct, manipulations. In one version of this effect, participants are presented with two conditions, one with low signal and low noise (low PE condition, Fig. 3a), and the other with high signal and high noise (high PE condition). In the other version of this effect, participants are presented with a two-choice task involving stimuli that contain some evidence in favor of both choices, and have to decide which choice has more evidence in favor of it. For example, in the conditions depicted in Fig. 3c, the task is to decide which of two superimposed digits (4 and 6 in this example) has a higher contrast. The high PE condition has both higher positive evidence (evidence in favor of the correct answer) than the low PE condition. In both versions of the effect, the PE bias manifests as higher confidence in the high vs. low PE conditions,

despite the fact that signal-to-noise ratio, and therefore decision accuracy, is balanced across these conditions. This bias is considered a key piece of evidence against the BE model (Fig. 1a), and in favor of the RCE model (Fig. 1b), since the BE model gives equal weight to the evidence both for and against a decision, whereas the RCE model considers only the evidence in favor of a decision.

Figure 3b and d show that both versions of the PE bias naturally emerged in our model across a range of conditions. For both the MNIST and CIFAR-10 datasets, confidence was higher in the high vs. low PE conditions, despite balanced accuracy, as previously observed in studies of human decision confidence[11,12,17]. The presence of this bias therefore did not depend on the specific dataset used, or the architectural details of the model, since experiments on CIFAR-10 used a more complex ResNet architecture for the encoder. In the orientation discrimination RL task, the opt-out rate was lower in the high vs. low PE

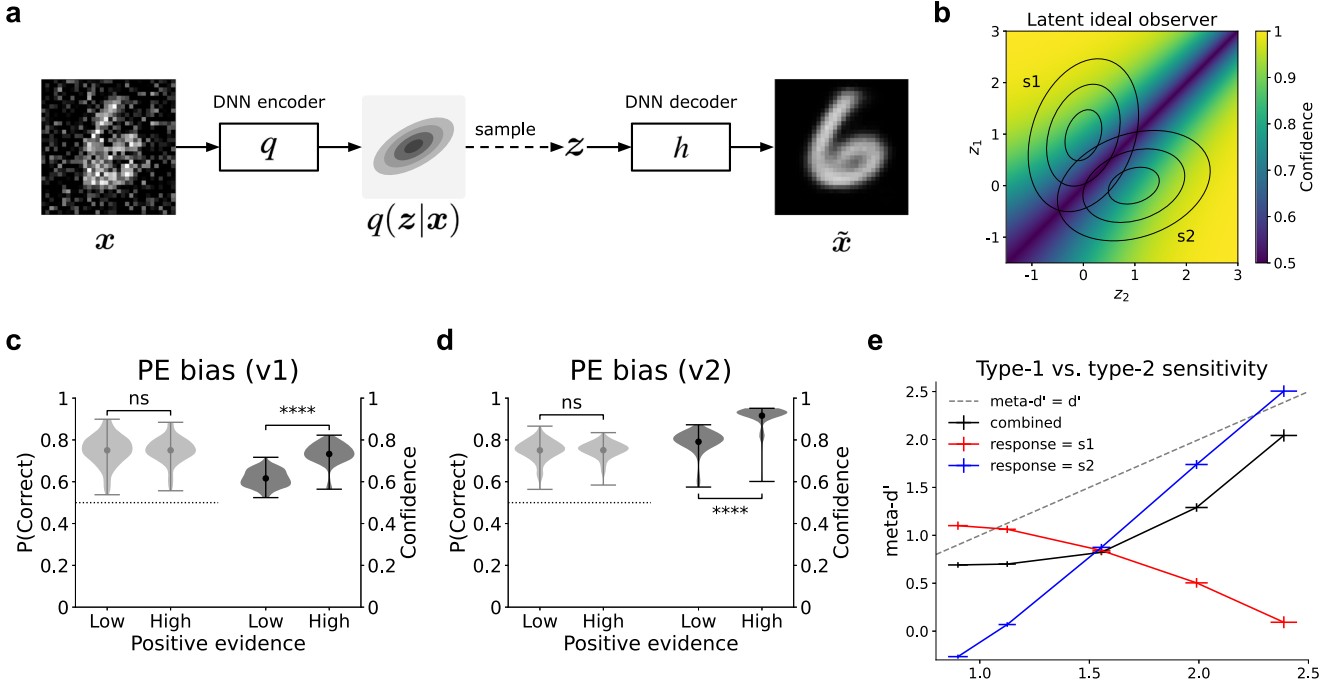

**Fig. 4 | Latent ideal observer accounts for dissociations between decisions and confidence. a** Denoising variational autoencoder (VAE) used to extract low-dimensional latent representation of training data. An image **x** was passed through a DNN encoder $q$ (distinct from the encoder $f$ used in the supervised neural network model), which output the parameters (means and variances) of the latent posterior $q(\mathbf{z}|\mathbf{x})$, a two-dimensional Gaussian distribution. This distribution was sampled from, yielding **z**, which was then passed through a DNN decoder $h$, yielding $\bar{\mathbf{x}}$, a denoised reconstruction of the input **x**. The VAE was regularized based on the divergence of the latent posterior from a unit normal prior (with means equal to 0 and variances equal to 1), encouraging efficient low-dimensional encodings of the high-dimensional inputs. **b** Latent ideal observer model. After training the VAE, Gaussian distributions were fit to the latent representations resulting from the training images for classes s1 and s2. The distributions were used to construct an ideal observer model that computed confidence according to $p(correct|\mathbf{z})$, the probability of being correct given the low-dimensional embedding **z**. Concentric

ellipses represent distributions based on the average parameters of those extracted from 100 trained VAEs. The latent ideal observer accounted for both versions of the PE bias, including (**c**) the version involving manipulation of contrast and noise (Fig. 3a; two-sided paired $t$-tests, accuracy: $p = 0.97$, confidence: $p = 1.3 \times 10^{-60}$), and (**d**) the version involving superimposed stimuli presented at different contrast levels (Fig. 3c; two-sided paired t-tests, accuracy: $p = 0.82$, confidence: $p = 1.3 \times 10^{-62}$). Confidence for correct and incorrect trials is shown in Supplementary Fig. S3. **e** The latent ideal observer also accounted for the dissociation between type-1 and type-2 sensitivity. Results in (**c**) and (**d**) reflect probability density over 100 ideal observers (each based on distributions extracted by a separate trained VAE), with mean accuracy/confidence in each condition represented by circular markers, and maxima/minima represented by the upper/lower line; Results in (**e**) reflect mean d'/meta-d' over 100 ideal observers ± the standard of the mean; dotted black lines represent chance performance; ns indicates $p > 0.05$, ****$p < 0.0001$. Source data are provided as a Source Data file.

conditions, as previously observed in studies using animal models[14,16]. The presence of this bias therefore did not depend on the use of supervised learning to train the confidence layer, but also emerged when using a more realistic training signal (reward).

**Dissociation between type-1 and type-2 sensitivity.** Maniscalco et al.[13] identified and confirmed a more specific, and surprising, prediction of the RCE model: human confidence ratings are, under certain conditions, characterized by a pattern of increasing type-1 sensitivity (as measured by decision accuracy or d') and decreasing type-2 sensitivity (as measured by meta-d'[20]). That is, confidence ratings become less diagnostic of decision accuracy as decision accuracy increases. The RCE model predicts that this pattern should emerge whenever a discrimination is made between two stimulus classes, one of which (s1) is presented at a fixed contrast, and one of which (s2) is presented at a variable contrast (Fig. 3e). Under these conditions, meta-d' increases as a function of d' for trials in which participants respond s2, and decreases as a function of d' for trials in which participants respond s1, resulting in the crossover pattern depicted in Fig. 3f (Behavior). This pattern is at odds with the BE model, according to which meta-d' should be equal to d'.

We simulated this paradigm in our model using a two-choice variant of the MNIST dataset, in which each model was trained to discriminate between two stimulus classes (e.g., 3 vs. 9). To account for the additional accumulation of noise between the time at which

decisions and confidence ratings are made, an additional noise parameter was added to the output of the network's confidence layer. The model showed a strikingly similar pattern to the previously observed results (Fig. 3f, Model), capturing both the crossover effect and the pattern of decreasing meta-d' as a function of increasing d' for trials with an s1 response. Furthermore, even without the addition of the noise parameter, these qualitative effects were still present (Supplementary Fig. S2).

**Latent ideal observer**

The previous results show that our model captures a number of established behavioral dissociations between confidence and decision accuracy. How can these dissociations be explained? One possibility is that, despite being extensively optimized to estimate confidence by predicting its own probability of being correct, the model nevertheless converged on a suboptimal heuristic strategy. An alternative possibility is that these effects reflect a strategy that is optimal given the actual distribution of the data for which the model was optimized, which may violate the assumptions underlying the presumed optimality of the BE rule. We found strong evidence to support this latter interpretation.

To answer this question, we first sought to quantitatively characterize the distribution of the model's training data. Because it is not tractable to perform ideal observer analysis directly on

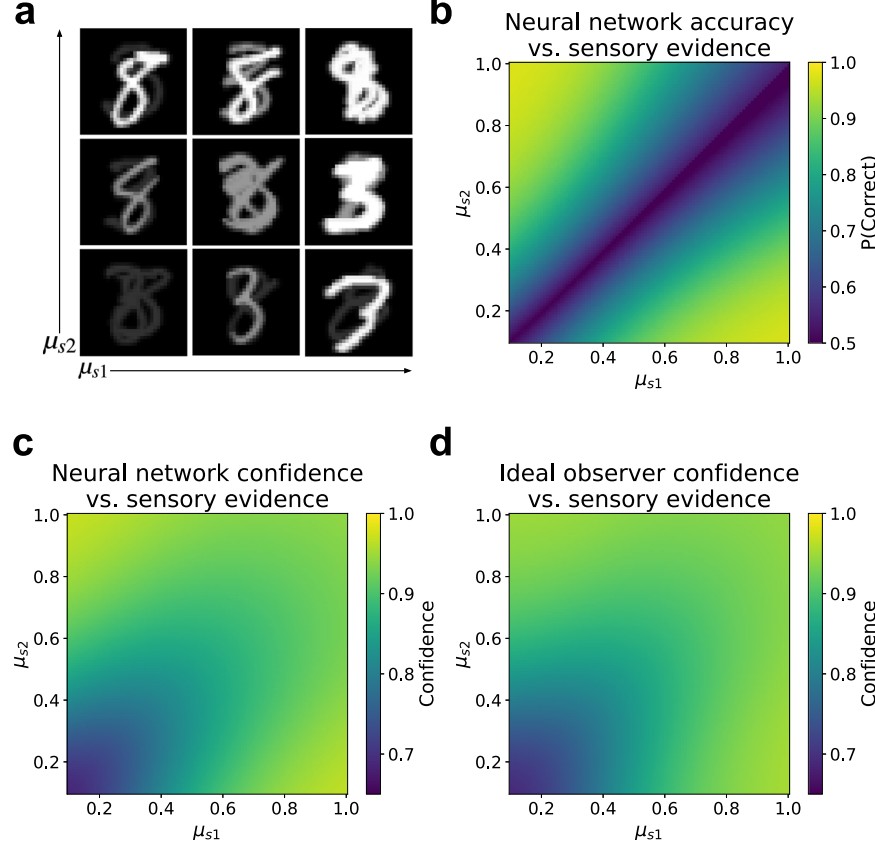

**Fig. 5 | Learned confidence strategy best explained by ideal observer.**
**a** Generalized version of positive evidence manipulation used to comprehensively evaluate both accuracy and confidence as a function of sensory evidence. Model was trained on classification of individual stimuli over the standard range of contrast and noise levels, then tested on images consisting of two superimposed stimuli belonging to classes s1 and s2, with independently varying contrast levels $\mu_{s1}$

and $\mu_{s2}$ (visual noise was also included in images presented to the model).
**b** Decision accuracy resembled the BE rule, as expected given uniform sampling of sensory evidence space. **c** Confidence displayed a more complex pattern.
**d** Confidence was best predicted by the latent ideal observer model, which outperformed regression models based on either the RCE or BE rules. Results reflect an average over 100 trained networks. Source data are provided as a Source Data file.

the model's high-dimensional inputs, we instead used unsupervised deep learning techniques to extract the low-dimensional, latent space underlying those inputs. Specifically, we used a denoising variational autoencoder (VAE; Fig. 4a)[21], which was trained to map a high-dimensional input **x** to a low-dimensional embedding **z** (consisting of just two dimensions), such that a denoised version of **x** can be decoded from **z**. Figure 4b depicts a summary of the low-dimensional latent distributions extracted by the VAE. These distributions had two important properties. First, these distributions had an elliptical shape, as quantified by the ratio of the variance along the major and minor axes ($\sigma_{target}/\sigma_{nontarget} = 2.44 \pm 0.04$ over 100 trained VAEs). Second, the distributions underlying classes s1 and s2 fell along non-parallel axes ($\theta_{s1,s2} = 51.4° \pm 1.7$). Under these conditions, the optimal approach for estimating confidence follows a more complex function than either the BE or RCE rules, as visualized in Fig. 4b. We then constructed an ideal observer that computed confidence according to $p(correct|\mathbf{z})$, using the distribution of the training data in the low-dimensional space extracted by the VAE, and we evaluated this function according to the distribution of the test data in this space. This ideal observer model robustly captured both versions of the PE bias (Fig. 4c, d), as well the dissociation between type-1 and type-2 sensitivity (Fig. 4e), thus replicating the same dissociations displayed by our performance-optimized neural network model. A more comprehensive analysis is presented in Supplementary Figs. S4 and S5, showing that the emergence of these biases depends on sensory evidence

distributions that are both asymmetrical ($\sigma_{target}/\sigma_{nontarget} > 1$), and non-parallel (as quantified by $0° < \theta_{s1,s2} < 180°$).

Importantly, we found that a key driver of this variance structure was the presence of variable contrast in the model's training data. When the training data involved only images presented at a fixed contrast, the distributions extracted by the VAE were characterized by asymmetric variance ($\sigma_{target}/\sigma_{nontarget} = 2.57 \pm 0.06$), but with a much smaller angular difference ($\theta_{s1,s2} = 10.8° \pm 1.6$, two-sample t-test, standard training regime vs. fixed-contrast regime, $t = 17.2$, $p < 0.0001$). In line with this observation, we show in "Dissociations are driven by statistics of training data" that manipulating this feature of the training data has a dramatic effect on the biases displayed by the model.

**Comparing the latent ideal observer and RCE models.** Given that the observed behavioral dissociations can, in principle, be explained both by the RCE heuristic model and our latent ideal observer model, we next sought to determine which of these models best characterized the behavior of the neural network. To do so, we employed a generalized version of the positive evidence manipulation, evaluating the neural network model across a grid of conditions, each of which was defined by a particular set of contrast levels for stimulus classes s1 and s2 (Fig. 5a). This allowed for a more comprehensive characterization of the model's behavior as a function of the sensory evidence space.

We found that the model's decision accuracy strongly resembled the BE rule (Fig. 5b), whereas confidence displayed a more complex pattern (Fig. 5c). To better understand the model's confidence behavior, we formally compared the pattern displayed in Fig. 5c to

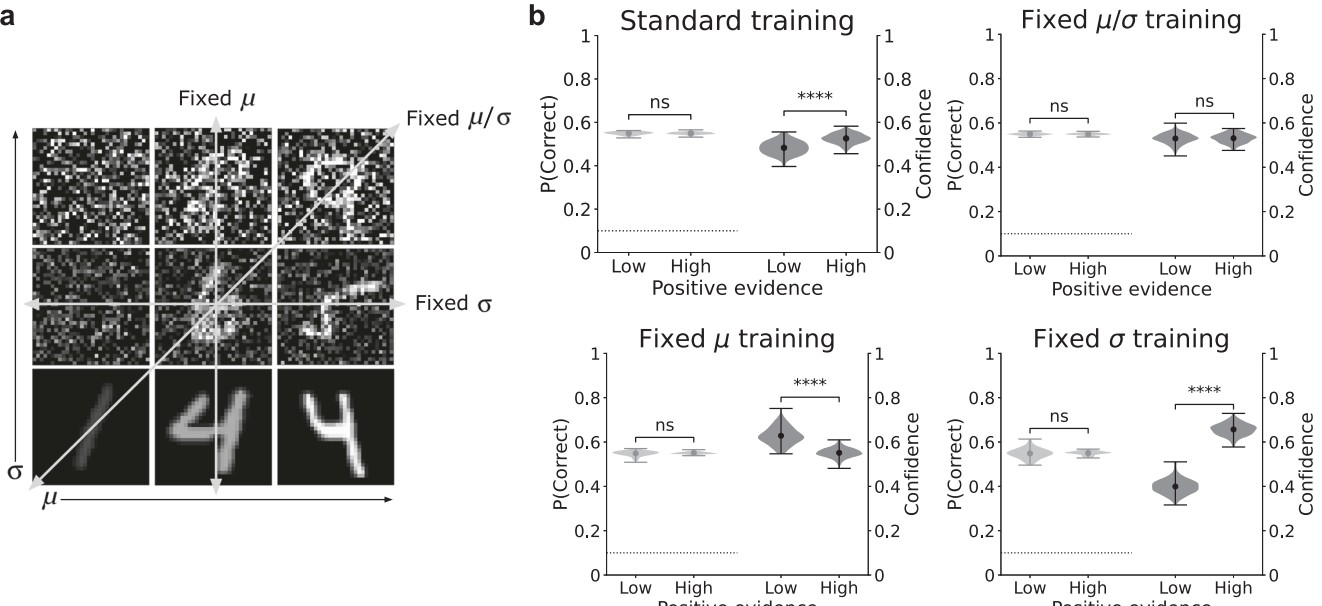

**Fig. 6 | The PE bias is driven by distribution of training data. a** Illustration of four different training regimes. In the standard training regime, the model was trained on the full range of variability in both contrast and noise levels. In the fixed $\mu/\sigma$ regime, the model was trained on images with a fixed signal-to-noise ratio, equivalent to being trained directly on the positive evidence manipulation. In the fixed $\mu$ regime, the model was trained on images that varied only in their noise level. In the fixed $\sigma$ regime, the model was trained on images that varied only in their contrast level. **b** Effect of training regime on presence of PE bias. Variance in both contrast and noise (standard training) resulted in PE bias (two-sided paired $t$-tests, accuracy: $p = 0.42$, confidence: $p = 1.4 \times 10^{-20}$). Fixed contrast-to-noise ratio (fixed $\mu/\sigma$ training) resulted in no PE bias (two-sided paired $t$-tests, accuracy: $p = 0.7$,

confidence: $p = 0.31$). Variance in noise only (fixed $\mu$ training) resulted in reversed PE bias—higher confidence in low vs. high PE conditions (two-sided paired $t$-tests, accuracy: $p = 0.21$, confidence: $p = 4.8 \times 10^{-35}$). Variance in contrast only (fixed $\sigma$ training) resulted in significantly larger PE bias than standard training regime (two-sided paired $t$-tests, accuracy: $p = 0.47$, confidence: $p = 4.8 \times 10^{-79}$). Results reflect probability density over 100 trained networks, with mean accuracy/confidence in each condition represented by circular markers, and maxima/minima represented by the upper/lower line; ns indicates $p > 0.05$, ****$p < 0.0001$. Confidence for correct and incorrect trials is shown in Supplementary Fig. S6. Source data are provided as a Source Data file.

regression models based on the BE and RCE rules, as well as a regression model based on the latent ideal observer. Confidence was better explained by the RCE rule than the BE rule (RCE $R^2 = 0.82 \pm 0.01$; BE $R^2 = 0.42 \pm 0.01$; paired t-test, RCE vs. BE, $t = 33.37$, $p < 0.0001$), but the ideal observer explained confidence better than either of these rules (ideal observer $R^2 = 0.89 \pm 0.01$, Fig. 5d; paired t-test, ideal observer vs. RCE, $t = 28.4$, $p < 0.0001$), and indeed was very close to the noise ceiling (the ability of the average pattern across networks to predict the behavior of individual networks, $R^2 = 0.9 \pm 0.01$).

These results give rise to a few questions. First, given that the ideal observer estimates confidence according to an optimal prediction of its own accuracy, what explains the difference between the patterns displayed by accuracy and confidence? This can be explained by the fact that the ideal observer's confidence estimates are based on the probability of being correct given the training distribution, which deviates from the distribution of the test conditions evaluated here. In particular, the conditions employed in this evaluation are designed to uniformly sample the sensory evidence space, whereas the training data are not uniformly distributed in this space. Second, why does the confidence pattern displayed by the ideal observer (Fig. 5d) differ from $p(\text{correct}|\mathbf{z})$, the function it uses to compute confidence (Fig. 4b)? This is because the test conditions depicted in Fig. 5a each contain their own degree of noise, and do not correspond to precise point estimates of the ideal observer's confidence function. Thus, the pattern displayed in Fig. 5d reflects essentially a smoothed version of the ideal observer model. Surprisingly, the result bears a strong visual resemblance to the RCE rule, though our quantitative analysis reveals that these two models can be distinguished, and that the ideal observer ultimately provides a better explanation of the confidence strategy learned by the neural network.

## Dissociations are driven by statistics of training data

The results of the ideal observer analysis suggest that the confidence biases displayed by the model—previously viewed as evidence of a suboptimal heuristic strategy—can be explained instead as arising from a strategy that is optimal given the distribution of the model's training data. One implication of this view is that a change to this distribution should lead to a change in the resulting biases.

To test this hypothesis, we studied how the PE bias was affected by the distribution of contrast and noise levels in the training data (Fig. 6a). Under the standard training regime, involving variation in both contrast and noise levels, the model displayed a PE bias (Fig. 6b, Standard training), but we found that this bias could be eliminated, reversed, or significantly enhanced, by altering this standard regime. Networks trained on data with a fixed contrast-to-noise ratio (Fixed $\mu/\sigma$ training), equivalent to being trained directly on the low and high PE conditions in which decision accuracy is balanced, did not display any bias at all. Networks trained under a regime in which contrast was fixed and only noise varied (Fixed $\mu$ training), displayed a reversed PE bias. That is, confidence was higher in the low vs. high PE conditions. This can be explained by the fact that, under this training regime, accuracy is primarily a function of the noise level, so it makes sense to adopt a confidence strategy based primarily on the level of sensory noise, resulting in higher confidence in the low PE (low noise) condition. Finally, networks trained under a regime in which noise was fixed and only contrast varied (Fixed $\sigma$ training), displayed a much larger PE bias, approximately five times as large as in the standard training regime (average confidence difference between the low and high PE conditions of $0.27 \pm 0.005$ in the fixed $\sigma$ regime vs. $0.05 \pm 0.004$ in the standard regime), consistent with the highly predictive relationship between contrast and accuracy in this training regime.

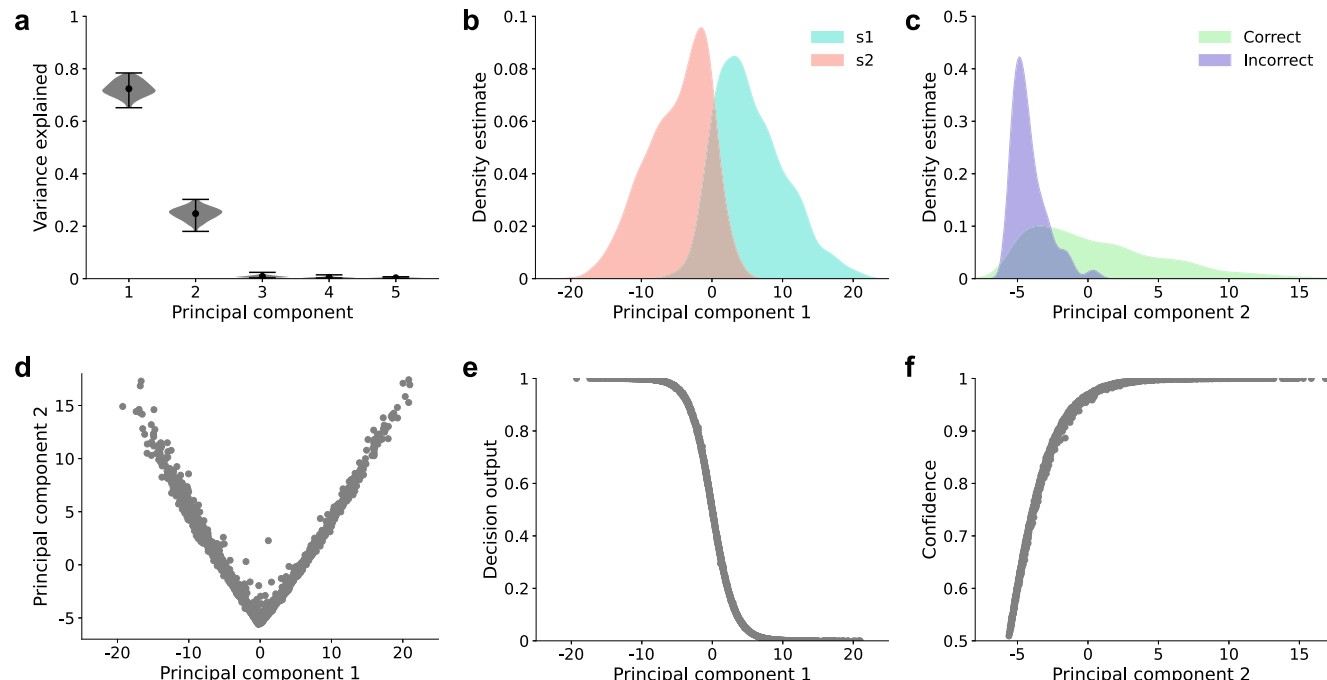

**Fig. 7 | Learned representations extract a common variable for both decisions and confidence. a** Principal component analysis revealed that the learned representations in the model's penultimate layer were almost entirely explained by the top two principal components. **a** Reflects probability density over 100 trained networks, with the mean represented by circular markers, and maxima/minima represented by the upper/lower line. **b**–**f** Depict results from a single trained network for the purposes of illustrating the model's learned two-dimensional geometry, but this general representational scheme was shared by all networks. **b** Kernel density estimates show that the distributions for s1 vs. s2 were separated along PC1. **c** The distributions for correct vs. incorrect trials were separated along PC2. **d** PC2 closely resembled a rectification of PC1. **e** PC1 predicted the model's decisions. **f** PC2 predicted the model's confidence ratings. Note that (**d**–**f**) depict trial-by-trial data points, not the predictions of a regression model. Source data are provided as a Source Data file.

Furthermore, the confidence strategy learned in the context of each training regime was generally the best strategy for that particular regime, in the sense that, for any given test regime, meta-d' was highest for models trained on that regime (Supplementary Fig. S7 and Table S1). Thus, the biases exhibited by the model under different training conditions can be viewed as the result of a confidence strategy that is best suited to a particular statistical regime. In particular, the presence of variable contrast in the model's training data appears to be a critical factor governing the emergence of a human-like bias toward positive evidence.

### Behavioral dissociations are consistent with common internal decision variable

An additional implication of the ideal observer model is that, despite the presence of dissociations at the behavioral level, decisions and confidence should nevertheless be based on a common internal decision variable. To investigate whether this was the case for our neural network model, we performed principal component analysis on the representations in the penultimate layer (the output of the encoder $f$ in Fig. 2a) of networks trained on the two-choice variant of MNIST. We found that the model's learned representations could be characterized by a two-dimensional geometry involving only a single decision variable. The top two principal components accounted for >97% of the variance (Fig. 7a). Principal component 1 (PC1) predicted both the stimulus class s1 vs. s2 (Fig. 7b; prediction accuracy = 0.88 ± 0.003), and the network's decision output (Fig. 7e; $R^2 = 0.99 ± 0.001$). Principal component 2 (PC2) predicted both whether a decision was correct vs. incorrect (Fig. 7c; prediction accuracy = 0.88 ± 0.003), and the network's confidence output (Fig. 7f; $R^2 = 0.9 ± 0.01$). Most importantly, PC2 corresponded closely to a rectification of PC1 (i.e., PC2 ∝ |PC1|; Fig. 7d; $R^2 = 0.86 ± 0.014$), suggesting that these components represented essentially the same decision variable. Indeed, the

same confidence biases displayed by the model were also exhibited by a rectified version of PC1, which the model used to make decisions, and even by a rectified version of the model's decision output itself (Supplementary Figs. S8–S10), consistent with the use of a common decision variable as in the ideal observer.

**Analysis of single unit representations.** The representational scheme learned by the model, in which both confidence and decisions are represented using a common decision variable, is reminiscent of findings at the single neuron level in the lateral intraparietal cortex (LIP)[3,5]. In those studies, nonhuman primates were presented with a perceptual decision-making task in which they sometimes had the option to opt out of the decision and receive a small but guaranteed reward (referred to as the sure target, or $T_S$), similar to the task that we used when training our model using RL. It was found that LIP neurons were both predictive of decisions, and implicitly encoded confidence, in the sense that they were more active for trials on which the neuron's preferred stimulus ($T_{in}$) was chosen vs. trials on which the sure target ($T_S$) was chosen. Similarly, these neurons were less active when their non-preferred stimulus ($T_{opp}$) was chosen. These neurons thus encoded both decisions and (implicitly) confidence as a single decision variable.

We tested whether our model would show similar effects, by analyzing responses at the single neuron level in the version of the model trained on the RL orientation discrimination task. Specifically, we analyzed the response of individual neurons in the penultimate layer of the network, which showed the same population-level representational signatures as the version of the model trained with supervised learning (i.e., a two-dimensional geometry representing a single decision variable; Supplementary Fig. S11). We evaluated the extent to which individual neurons were predictive of either the model's decision output (quantified by $R^2_{decision}$), or the opt-out output

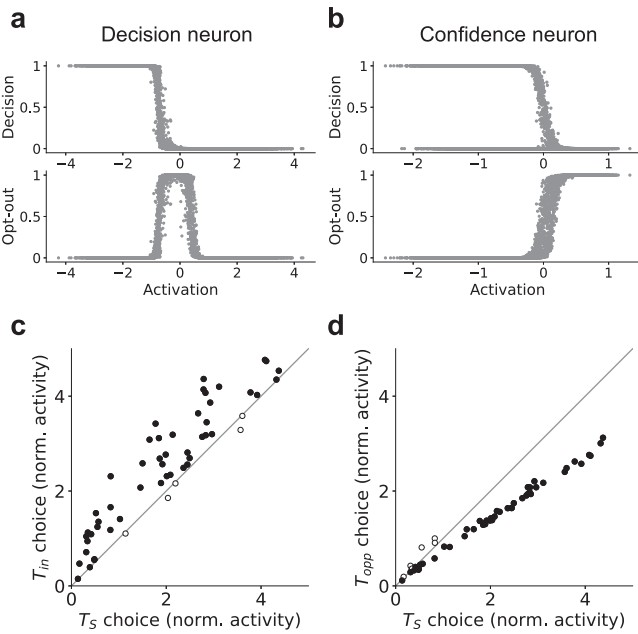

**Fig. 8 | Analysis of single unit representations. a** Example decision neuron, strongly predictive of decision output ($R^2_{decision} = 0.83$), but not opt-out response ($R^2_{opt-out} = 0.004$). **b** Example confidence neuron, strongly predictive of opt-out response ($R^2_{opt-out} = 0.87$), but not decision output ($R^2_{decision} = 0.16$). Kiani & Shadlen[3] found that decision-making neurons in the lateral intraparietal cortex (LIP) implicitly coded for confidence. Decision neurons in our neural network model showed this same pattern. **c** They showed higher activity for trials on which their preferred stimulus ($T_{in}$) was chosen vs. trials on which a sure target ($T_S$) was chosen, whereas (**d**) they were less active for trials on which their non-preferred stimulus ($T_{opp}$) was chosen. Each point in **c** and **d** represents the average normalized (against the pre-stimulus baseline) activation of an individual neuron (over $N \geq 2700$ trials) ± the standard error of the mean. Black dots represent neurons with statistically significant deviations from the diagonal (two-sided two-sample $t$-tests, $p < 0.05$). White dots represent neurons with non-significant deviations (two-sided two-sample $t$-tests, $p > 0.05$). Note that error bars are present on these plots, but are too small to be visible. Neural network results reflect a single example network, but all 100 trained networks displayed a qualitatively similar pattern. Source data are provided as a Source Data file.

(quantified by $R^2_{opt-out}$). We found that some neurons were strongly predictive of decisions (Fig. 8a), while other neurons were strongly predictive of the opt-out output (Fig. 8b). We classified neurons as either decision neurons or confidence neurons, by computing $\Delta R^2 = R^2_{decision} - R^2_{opt-out}$ (see "Single unit analysis" and Supplementary Fig. S12). The decision neurons in our model showed a pattern very similar to the behavior of LIP neurons: they were more active on trials where the preferred stimulus was chosen vs. opt-out trials (Fig. 8c), and were less active on trials where the non-preferred stimulus was chosen (Fig. 8d; paired $t$-tests, $p < 0.05$ for 97 out of 100 trained networks).

It should be noted that, although we treat individual neurons as belonging to discrete categories (decision vs. confidence neurons) in order to compare with previous results, a closer analysis suggests a more distributed pattern. This can be seen by comparing $\Delta R^2$ with the projection of each neuron onto the top 2 PCs (Supplementary Fig. S12). Neurons aligned with PC1 were strongly predictive of decisions, while neurons aligned with PC2 were strongly predictive of confidence. But there were also other neurons that interpolated between these clusters, such that they were moderately predictive of both decisions and confidence, and were not strongly aligned with either PC. Thus, although it is possible to view the individual neurons in our model as belonging to discrete categories, a more general interpretation is to view them as jointly representing a low-dimensional geometry at the population level.

## Accounting for neural dissociations

In the previous section, we showed that the model learned to use a single internal decision variable for both decisions and confidence, despite the observed dissociations at the behavioral level. However, a number of previous results have also reported dissociations between the neural processes underlying decisions vs. confidence. Here, we show that our neural network model can also account for many of these neural dissociations, demonstrating that, contrary to previous interpretations, these dissociations are consistent with the use of a common decision variable.

**Contribution of decoded neural evidence to confidence.** One such neural dissociation was reported by Peters et al.[15], who recorded whole-brain cortical electrophysiological signals while participants performed a face/house discrimination task, and trained a classifier to estimate the amount of neural evidence in favor of faces vs. houses on a trial-by-trial basis. This decoded neural evidence measure was then used to determine whether decisions and confidence were better predicted by the balance of evidence for faces vs. houses, or by the response-congruent evidence alone. Both receiver operating characteristic (ROC) and choice probability (the area under the ROC curve) analyses revealed that decisions were better predicted by the BE rule, whereas confidence was predicted about equally well by either the BE or RCE rules, meaning that the incorporation of decision-incongruent evidence did not significantly improve the prediction. These results seem to imply that decisions and confidence are based on distinct neural decision variables (BE for decisions, and RCE for confidence), at odds with the representation learned by our model.

We simulated this analysis in our model by training a separate decoder to predict the stimulus class s1 vs. s2 given the concatenated activation states of all layers in the network, mirroring the whole-brain decoding approach in the original study. We were surprised to find that, despite the use of a common decision variable represented in the penultimate layer of the network, this analysis produced a strikingly similar dissociation when decoding from all layers of the network (Supplementary Fig. S13). By contrast, when decoding only from the penultimate layer, both decisions and confidence were better predicted by the BE vs. RCE rules (Supplementary Figs. S14 and S15), consistent with the presence of a common decision variable. In principle, since the penultimate layer forms a part of the feature space in the whole-network decoding analyses, and decisions are nearly perfectly decodable from this layer, the optimal decoder should be able to ignore the earlier layers and produce the same results for both analyses. However, the very large number of features present in the whole-network analysis (11,236 neurons) necessitated the use of a decoding method based on gradient descent, which is susceptible to local minima, and therefore will not necessarily converge to the optimal result. This likely explains the discrepancy between these analyses.

These results demonstrate how such whole-brain decoding analyses may offer a misleading characterization of the high-level decision variables utilized by the brain. This may be especially true when analyzing data characterized by relatively low spatial resolution, given previous findings that such analyses are more sensitive to neural signals from earlier sensory regions than higher-order decision-making regions[22]. This suggests the need for direct intracellular recordings from decision-making areas, e.g., lateral intraparietal cortex (LIP) or dorsolateral prefrontal cortex (dlPFC), to test the predictions of the model.

**Dissociations resulting from brain stimulation.** A few studies have found that the application of transcranial magnetic stimulation (TMS) to specific brain regions can have dissociable effects on decisions and confidence. In one study, it was found that low intensity TMS to primary visual cortex (V1) led to a pattern of decreased type-1 sensitivity (d′) combined with increased confidence[23]. We simulated this

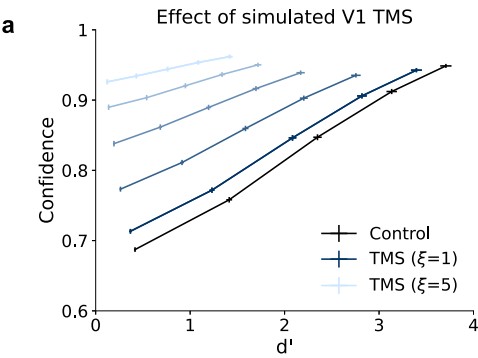
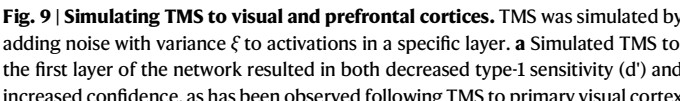
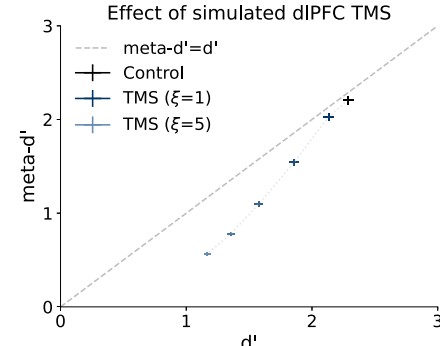

**Fig. 9 | Simulating TMS to visual and prefrontal cortices.** TMS was simulated by adding noise with variance $\xi$ to activations in a specific layer. **a** Simulated TMS to the first layer of the network resulted in both decreased type-1 sensitivity (d') and increased confidence, as has been observed following TMS to primary visual cortex (V1). **b** Simulated TMS to the penultimate layer of the network resulted in a selective impairment of type-2 sensitivity (meta-d'), as has been observed following TMS to dorsolateral prefrontal cortex (dlPFC). See Supplementary Fig. S16 for a detection-theoretic explanation of this effect. Source data are provided as a Source Data file.

experiment in our model by adding random noise (with variance $\xi$) to the activations in the first layer of the network, which captured the simultaneous decrease in d' and increase in confidence as a result of increasing TMS strength (Fig. 9a). Another study found that theta-burst TMS to dlPFC resulted in a greater impairment of type-2 sensitivity (meta-d') than type-1 sensitivity (d')[24]. We simulated this experiment in our model by adding random noise to the activations in the penultimate layer of the network, which captured the greater impact of TMS on meta-d' vs. d' (Fig. 9b). These results are both amenable to standard detection-theoretic explanations—an explanation of the effect of TMS to V1 was proposed in the original study[23], and we present an explanation of the effect of TMS to dlPFC in Supplementary Fig. S16. However, they highlight the ability of the neural network model to capture broad anatomical distinctions in the neural mechanisms underlying confidence.

**Blindsight.** One particularly striking dissociation between decisions and confidence comes from the condition known as blindsight[25], in which a lesion to V1 results in the subjective sensation of blindness despite relatively preserved visual capacity. One way to characterize blindsight is as a severe deficit in visual metacognition. Blindsight patients typically have very low confidence in the visual discriminations made in their blind hemifield, routinely referring to them as guesses[25]. Even in the rare cases where patients express high confidence in their visual discriminations (e.g., in the case of blindsight patient GY), their confidence ratings are generally not very predictive of whether those discriminations will be correct or incorrect[26,27]. This can be formalized as a pattern of relatively preserved d' combined with very low meta-d'.

We simulated lesions to V1 by scaling the activations in the first layer of the trained neural network model by a factor of 0.01. The small amount of signal remaining in the first layer of the network was intended to model the intact visual signals from subcortical regions that are thought to mediate residual visual function in blindsight[28]. We found that, despite this significant scaling of activations, the model retained substantial visual function, as indicated by high d' values, whereas the model's confidence ratings were no longer predictive of performance, as indicated by meta-d' ≈ 0 (Fig. 10a). This is in contrast to control networks, without a lesion, which displayed meta-d' ≈ d'. We note that, in our analysis, confidence criteria were selected based on the empirical distribution of confidence outputs for each condition, so meta-d' ≈ 0 cannot be trivially explained by confidence ratings that all fall below an arbitrary pre-lesion criterion. We also computed density estimates for confidence ratings on correct vs. incorrect trials, which further confirmed that, in lesioned networks, confidence ratings were not only low but also had completely overlapping distributions for—and thus could not discriminate between—correct and incorrect trials

(Fig. 10b). The model therefore captured the key metacognitive components of blindsight.

It should be noted that the exact extent of the metacognitive impairment in blindsight is currently unclear, with some data suggesting that meta-d' is significantly lower, though still above zero[26]. Our model can also account for this pattern (Supplementary Fig. S17). It should also be noted that the small negative values of meta-d' in Fig. 10a are most likely due to the highly non-Gaussian nature of the confidence distributions, which violates the assumptions of meta-d'.

**Prefrontal regions selectively involved in confidence.** A number of studies have found that impairment of specific brain regions—either through lesions or temporary inactivation—can selectively impair confidence, without affecting first-order judgments themselves[29–32]. We investigated whether these effects could be captured by simulating a lesion to the penultimate layer of the network, but did not find any such dissociation (Supplementary Fig. S18). This suggests the need for further model development to capture the role that specific brain regions play in confidence.

## Discussion

The question of whether confidence judgments reflect an optimal prediction of the probability of being correct has been hotly debated in recent years[3–9,11–19,33,34]. Those debates have centered around models that rely on strong assumptions about the representations over which confidence is computed. Previous work using artificial neural networks to model decision confidence has also generally relied on such representational assumptions[35–38]. In this work, by contrast, we used deep neural networks to study confidence in the context of realistic, high-dimensional stimuli. We found that a relatively simple model, optimized only to predict its own likelihood of being correct, captured many of the biases and dissociations that have driven recent debates. Furthermore, we found that an ideal observer applied to a low-dimensional projection of the model's training data yielded a rational explanation of these biases, and provided a very close fit to the pattern of confidence displayed by the model.

Our findings have an important link to models of decision confidence in which sensory evidence distributions are characterized by asymmetric variance[18,19], according to which the optimal confidence strategy follows a more complex function than a simple BE rule. We found that a low-dimensional projection of our model's training data displayed this key property of asymmetric variance, driven to a significant extent by the presence of variable contrast in the training data. Consistent with this, we found that the PE bias could be eliminated, or even reversed by manipulating this feature of the training data. More broadly, these results suggest that human confidence biases may emerge as a consequence of optimization for a particular statistical

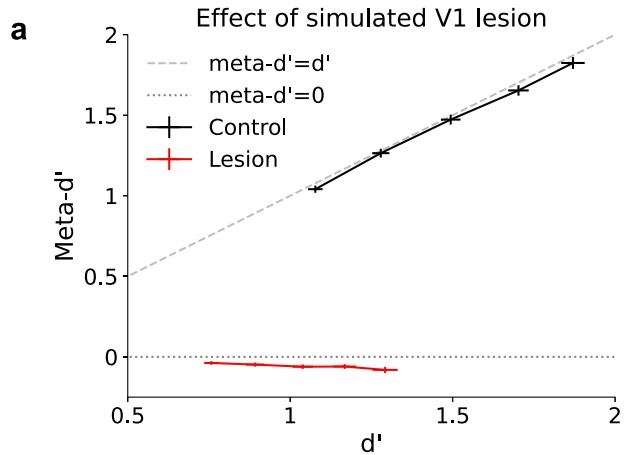

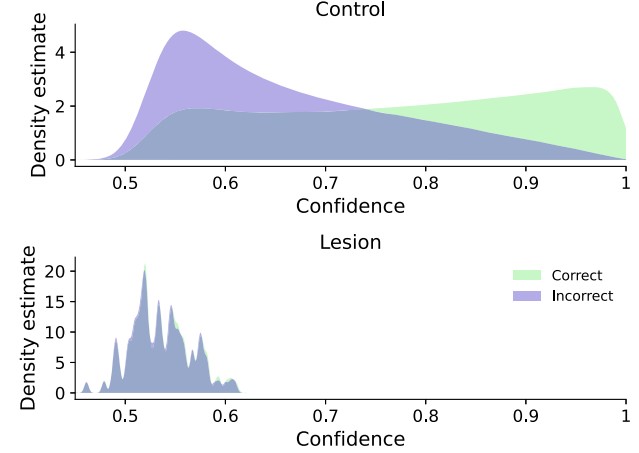

**Fig. 10 | Simulating blindsight.** Lesions to V1 can cause the condition known as blindsight, in which patients have the subjective sensation of blindness despite preserved visual capacity. This pattern can be formalized as a combination of preserved type-1 sensitivity (d'), low visual confidence, and low type-2 sensitivity (meta-d'). **a** Lesions to V1 were simulated in the model by scaling activations in the first layer of the trained network by a factor of 0.01. This resulted in a sharp reduction in meta-d' despite relatively preserved d'. **b** Confidence ratings were significantly lower following simulated lesions, and the distribution of confidence ratings showed a nearly complete overlap for correct and incorrect trials, consistent with meta-d' ≈ 0. All results reflect an average over 100 trained networks ± the standard error of the mean. Source data are provided as a Source Data file.

regime, in particular one governed by variable signal strength, and it therefore may be possible to reshape them through direct training under specific task conditions. This idea is consistent with the results of Maniscalco et al.[13], who found that the dissociation between type-1 and type-2 sensitivity could be largely eliminated when participants received feedback about the effectiveness of their type-2 responses.

Our model is also related to the recently proposed model of Khalvati et al.[34], in that both models propose a rational account of seemingly suboptimal dissociations between decisions and confidence. An important difference is that the model of Khalvati et al. depends on a unidimenionsal representation of sensory evidence, and therefore cannot account for effects that require a two-dimensional sensory evidence space. These include the dissociation between type-1 and type-2 sensitivity, and the version of the PE bias involving superimposed stimuli, both of which our model accounts for. However, a deep commonality is that both models invoke a distinction between the sensory evidence distribution assumed by the experimenter, and the actual distribution used by the decision-maker. In the case of our model, we specifically propose that such dissociations arise because they are globally optimal according to the broader distribution of the decision-maker's prior sensory experiences, even if they are not locally optimal for the sensory evidence distribution in a specific task. There is also an interesting parallel to models of confidence in memory in which fluency serves as a cue to the reliability of a memory[39]. There too, one possible explanation is that this heuristic is employed because fluency is generally a reliable predictor of the accuracy of one's memories[40], even if it is not necessarily diagnostic in every task setting[41].

We found that the representations learned by our model were characterized at the population level by a two-dimensional geometry, in which one dimension coded for decisions, and another dimension coded for confidence. At the single neuron level, we found that some neurons were both strongly predictive of decisions, and implicitly coded for confidence, in the sense that they responded more to their preferred stimulus class when the network displayed high confidence. These results mirror previous findings from decision-making neurons in LIP[3,5]. In addition to this implicit coding of confidence, we also found that other neurons explicitly coded for confidence, in the sense that confidence could be linearly decoded from their activation level, regardless of which choice was made by the network. Such explicit

representations of confidence have been discovered at the single neuron level in both orbitofrontal cortex[42] and the pulvinar[29].

A related finding is that a number of brain regions appear to play a selective role in confidence. Temporary inactivation of either orbitofrontal cortex[31] or pulvinar[29] affects confidence-related behaviors without affecting decisions; temporary inactivation of specific prefrontal nodes (areas 9 and 6) has dissociable effects on memory confidence, without affecting memory itself[32]; and lesions to anterior prefrontal cortex cause a domain-specific impairment of perceptual metacognitive accuracy, without impairing perceptual decision accuracy[30]. Our model did not capture this segregation of confidence vs. first-order decision-making capacities into distinct regions. Instead, confidence and decision neurons were intermixed in the penultimate layer of the network, and a simulated lesion to this layer did not cause a selective metacognitive impairment. One likely reason is that, unlike the brain[43,44], there is no pressure in our model for neurons with similar functionality to cluster together spatially. Additionally, our model is missing a number of architectural elements thought to be important for the function of these brain regions, including recurrence, top-down feedback connections, and convergent multimodal inputs. The incorporation of these elements is a promising avenue for future work.

An additional issue is the question of whether decisions and confidence are computed based on a common decision variable (which is distinct from the question of whether they are supported by different brain regions). It has recently been proposed that confidence and decisions are supported by distinct populations of neurons with differing degrees of lateral inhibition, such that decisions and confidence are based on different weightings of the sensory evidence[37]. This model was motivated by, and can account for, some of the same behavioral dissociations that we model in the present study. In contrast, our analyses of the neural network model's learned representations provided strong evidence that it relied on a common decision variable, and this conclusion is also consistent with the results of our ideal observer model, which employs a common decision variable by definition. A key contribution of the present study is thus to show how the previously observed dissociations can arise despite the use of a common decision variable. This also leads to the prediction that the neural decision variable underlying decisions should be subject to the same biases as confidence, even if decisions don't display those biases at a behavioral level.

We have presented a simple, high-level model that abstracts over many important properties of biological neural networks, including the presence of distinct cell types, temporal dynamics, etc. Though this level of abstraction has proven useful, it will also be important to expand the functionality and biological detail of the model in future work. For instance, the model could be expanded by replacing the feedforward encoder with a recurrent network[45], allowing the model to make contact with the rich body of data on temporal evidence accumulation in decision-making[46,47], and to account for phenomena such as the effect of post-decisional evidence on confidence judgments[48]. Furthermore, recently developed techniques[49] could be used to incorporate distinct inhibitory and excitatory cell types, allowing the model to implement lateral inhibition, which is thought to play a key role in the evidence accumulation process[50], and also figures prominently in mechanistic accounts of decision confidence[37]. The present approach therefore can be viewed as a general explanatory framework that suggests a number of exciting prospects for future work.

## Methods

### Code and hardware
Simulations were carried out in Python using the following packages: PyTorch[51], NumPy[52], SciPy[53], scikit-learn[54], and Matplotlib[55]. All code is available at: https://github.com/taylorwwebb/performance_optimized_NN_confidence

All simulations were performed using a single NVIDIA GeForce RTX 2080 Ti GPU.

### Datasets
Experiments were performed on three datasets. The first was the MNIST handwritten digits dataset, consisting of grayscale images of the digits 0-9[56]. This dataset has a training set with 60,000 images, and a test set with 10,000 images. The second was the CIFAR-10 object classification dataset, consisting of color images of 10 common object categories (cats, dogs, cars, etc.)[57]. This dataset has a training set with 50,000 images, and a test set with 10,000 images. Third, we used an orientation discrimination task featuring oriented gabor patches. Images had a size of 32x32, and each image contained a single gabor patch, tilted either 5 degrees to the left or 5 degrees to the right. Gabor patches were generated using a centered Gaussian envelope with a standard deviation of 4 pixels, and a spatial frequency of 0.3 cycles per pixel.

For experiments using the MNIST and CIFAR-10 datasets, all networks were trained and evaluated on the official training and test sets respectively. Thus, for these datasets, all results presented in the paper involve generalization beyond the images used for training. Images from the MNIST dataset were resized from 28x28 to 32x32 using bilinear interpolation. For training on the CIFAR-10 dataset (with an original image size of 32x32), a random crop of size 32x32 was selected after zero-padding of size 4 on all sides, and images were flipped horizontally with a probability of 0.5.

During training, images were scaled by a contrast factor $\mu$. This value was sampled online from a uniform distribution, and then multiplied by the values of each pixel (which had an original range of [0,1]). Images were then normalized to the range $[-1,1]$, and pixelwise noise was added. Noise was sampled from a Gaussian distribution with variance $\sigma$. This value was also sampled online from a uniform distribution. After adding noise, images were thresholded to the range $[-1,1]$ using a hard tanh function. For training on CIFAR-10, contrast was sampled from the range $[\mu=0.1, \mu=1]$ and noise was sampled from the range $[\sigma=0.1, \sigma=0.2]$. For training on the orientation discrimination task, contrast was sampled from the range $[\mu=0.1, \mu=1]$ and noise was sampled from the range $[\sigma=0.5, \sigma=1]$. For the standard MNIST training regime, contrast was sampled from the range $[\mu=0.1, \mu=1]$ and noise was sampled from the range $[\sigma=1, \sigma=2]$. Some experiments on MNIST used alternative training regimes, as described in "Alternative training and test regimes".

### Model architecture
The model architecture involved three major components. The first is a DNN encoder $f$ that takes an image $\mathbf{x}$, of class $y$. The output of this encoder is then passed to two output layers $g_{class}$ and $g_{conf}$. The layer $g_{class}$ outputs a predicted class $\hat{y}$ for the image:

$$\hat{y} = g_{class}(f(\mathbf{x})) \tag{1}$$

and the layer $g_{conf}$ predicts $p(\hat{y}=y)$, the probability that the classification response is correct:

$$p(\hat{y}=y) = g_{conf}(f(\mathbf{x})) \tag{2}$$

The architectural details of these components depended on the specific datasets and experiments.

**Encoder.** For both handwritten digit classification (using the MNIST dataset) and orientation discrimination (trained with RL), $f$ consisted of 3 convolutional layers followed by 3 fully-connected (FC) layers. The convolutional layers had 32 channels each, a kernel size of 3, a stride of 2, batch normalization, and leaky ReLU nonlinearities with a negative slope of 0.01. The first 2 FC layers had 256 and 128 units, batch normalization, and leaky ReLU nonlinearities with a negative slope of 0.01. The output of the encoder was generated by the final FC layer, which had 100 units and no nonlinearity or normalization.

For experiments on CIFAR-10, a more challenging object classification benchmark, $f$ employed a more complex ResNet architecture modeled closely on He et al.[58]. The basic building block of this architecture is the residual block, in which the input to a series of convolutional layers is added to their output, thus providing shortcuts in the computational graph that facilitate learning in very deep architectures. Our implementation of this component used the following formulation:

$$\tilde{\mathbf{x}}_b^{l=1} = \text{ReLU}\left(\text{BN}_b^{l=1}\left(\text{conv}_b^{l=1}(\mathbf{x}_{b-1})\right)\right) \tag{3}$$

$$\tilde{\mathbf{x}}_b^{l=2} = \text{BN}_b^{l=2}\left(\text{conv}_b^{l=2}(\tilde{\mathbf{x}}_b^{l=1})\right) \tag{4}$$

$$\mathbf{x}_b = \text{ReLU}\left(\tilde{\mathbf{x}}_b^{l=2} + \mathbf{x}_{b-1}\right) \tag{5}$$

where $\mathbf{x}_{b-1}$ is the output of the previous residual block $b-1$, $\text{conv}_b^{l=1}$ and $\text{conv}_b^{l=2}$ are the first and second convolutional layers in the current block $b$, and $\text{BN}_b^{l=1}$ and $\text{BN}_b^{l=2}$ are batch normalization layers. All convolutional layers employed a kernel size of 3 and no bias term. The default residual block employed layers with a stride of 1, and the same number of output channels as input channels. In some cases (as detailed in the next paragraph), a residual block had a different number of output than input channels, or employed a stride of 2 so as to output a smaller feature map. In these cases, the first convolutional layer $\text{conv}_b^{l=1}$ implemented either the stride or change in number of channels, and $\mathbf{x}_{b-1}$ was passed through an additional convolutional layer (incorporating the stride or change in number of channels) and batch normalization layer before being added to $\tilde{\mathbf{x}}_b^{l=2}$, to ensure that they had the same shape:

$$\tilde{\mathbf{x}}_b^{l=3} = \text{BN}_b^{l=3}\left(\text{conv}_b^{l=3}(\mathbf{x}_{b-1})\right) \tag{6}$$

$$\mathbf{x}_b = \text{ReLU}\left(\tilde{\mathbf{x}}_b^{l=2} + \tilde{\mathbf{x}}_b^{l=3}\right) \tag{7}$$

Residual blocks were further arranged into stacks. Each stack contained 9 residual blocks. Some stacks had a different number of output than input channels, implemented by the first block in the stack, and some stacks had an output stride of 2, implemented by the last block in the stack. The input image was first passed through an initial convolutional layer, with 16 channels, a stride of 1, batch normalization, and ReLU nonlinearities. This was followed by 3 residual stacks. The first stack had 16 output channels and an output stride of 2. The second stack had 32 output channels and an output stride of 2. The third stack had 64 output channels and an output stride of 1. Altogether, $f$ had 55 layers (1 initial layer + 3 stacks × 9 blocks per stack × 2 layers per block). Finally, the output of the encoder was generated by average pooling over the output of the third stack, yielding a 64-dimensional vector.

**Output layers.** For experiments on the standard versions of CIFAR-10 and MNIST (both 10-way classification tasks), $g_{class}$ parameterized a categorical distribution indicating the predicted probability that $\mathbf{x}$ belonged to each of the 10 possible image classes, using a linear layer followed by a softmax nonlinearity. For the two-choice versions of MNIST and CIFAR-10 (classification of two randomly selected classes s1 and s2), $g_{class}$ parameterized a binomial distribution indicating the predicted probability that $\mathbf{x}$ belonged to class s2, using a linear layer followed by a sigmoid nonlinearity. For all experiments on CIFAR-10 and MNIST, $g_{conf}$ parameterized a binomial distribution indicating the predicted probability that the classification response was correct.

For the orientation discrimination task, we used an actor-critic architecture trained with RL. The actor and critic were separate output layers that both took the output of the encoder $f(\mathbf{x})$ as input. The actor parameterized a categorical distribution over 3 possible actions (LEFT, RIGHT, and OPT-OUT) using a linear layer followed by a softmax nonlinearity. The critic predicted the reward that would be received on the current trial, using a linear layer.

### Training

**MNIST.** For the standard 10-choice version of MNIST, $g_{class}$ was trained with a cross-entropy loss over the 10 possible image classes. For the two-choice version of MNIST, $g_{class}$ was trained with a binary cross-entropy loss. The target was 0 if $\mathbf{x}$ belonged to class s1 and 1 if $\mathbf{x}$ belonged to class s2. The confidence output layer $g_{conf}$ was trained with a binary cross-entropy loss. The target was 1 if the classification output was correct and 0 if the classification output was incorrect. The classification and confidence losses were summed, and the entire architecture was trained through backpropagation. Training was performed for 5 epochs using the Adam optimizer[59], with a learning rate of $5e-4$ and a batch size of 32. Note that for the two-choice version, these epochs were about 1/5 as long as they were for the standard version (since they only involved 2 out of the 10 possible image classes). All weights and biases were initialized using PyTorch defaults.

**CIFAR-10.** For the CIFAR-10 dataset, $g_{class}$ and $g_{conf}$ were trained with the same loss functions used for MNIST (either cross-entropy loss (10-choice) or binary cross-entropy loss (two-choice) for classification, binary cross-entropy loss for confidence), which were summed and used to train the entire architecture through backpropagation. Networks were trained for 164 epochs using stochastic gradient descent with weight decay of $1e-4$, momentum of 0.9, and a batch size of 128. An initial learning rate of 0.1 was used, which was then set to 0.01 at training epoch 82, and 0.001 at training epoch 123. All weights in $f$ were initialized using a Kaiming normal distribution[60], and all weights in the output layers $g_{class}$ and $g_{conf}$ were initialized using an Xavier normal distribution[61].

**Reinforcement learning.** For the orientation discrimination task, networks were trained using an actor-critic method[62]. During training,

an action $a_t$ was selected on each trial by sampling from the probability distribution generated by the actor. If LEFT or RIGHT was selected, the reward $r_t$ received on that trial was 1 if the decision was correct and 0 if the decision was incorrect. If OPT-OUT was selected, the network received a smaller but guaranteed reward $r_{opt-out}$. This value was initialized to 0.5 at the beginning of training, and was then updated after each training batch $i$ according to the following formula:

$$r_{opt-out_i} = \min(p(\text{correct})_{i-1}, 0.75) \tag{8}$$

where $p(\text{correct})_{i-1}$ is the average accuracy for non-opt-out trials on the previous training batch. This setup prevented networks from defaulting to a strategy of always opting out early in training when accuracy was low.

Networks were trained to maximize reward in this task using the sum of two loss functions. The critic was trained using a smooth L1 loss to generate $v_t$, a prediction of the reward $r_t$ received on that trial. This was then used to compute a reward prediction error:

$$\delta_t = r_t - v_t \tag{9}$$

which was used to compute a loss function for training the actor:

$$L_{actor} = -\log(p_{a_t})\delta_t \tag{10}$$

where $-\log(p_{a_t})$ is the negative log likelihood of the action sampled on that trial. The actor and critic losses were summed, and the entire architecture was trained through backpropagation. Networks were trained for 5000 iterations, using the Adam optimizer, with a learning rate of 0.001 and a batch size of 32. All weights and biases were initialized using PyTorch defaults.

### Latent ideal observer

**Variational autoencoder.** We used a denoising variational autoencoder (VAE)[21] to learn a low-dimensional representation of the neural network model's training data. Like a standard autoencoder, the VAE involves a neural network encoder that maps a high-dimensional input $\mathbf{x}$ to a low-dimensional embedding $\mathbf{z}$ (consisting of just 2 dimensions in our case), and a neural network decoder that is trained to reconstruct $\mathbf{x}$ given $\mathbf{z}$. However, unlike a standard autoencoder, which produces a deterministic latent embedding, the VAE maps each input $\mathbf{x}$ to a latent distribution $q(\mathbf{z}|\mathbf{x})$, by parameterizing the means and variances of this distribution using separate output layers from the encoder. This distribution is then sampled from yielding $\mathbf{z} \sim q(\mathbf{z}|\mathbf{x})$, which is passed to the decoder. Importantly, in addition to being trained with a standard reconstruction objective, the latent representations in the VAE are regularized according the $D_{KL}(q(\mathbf{z}|\mathbf{x}), p(\mathbf{z}))$, the Kullback-Leibler divergence between $q(\mathbf{z}|\mathbf{x})$ and $p(\mathbf{z})$, a unit normal distribution with $\mu = 0$ and $\sigma = 1$. This regularization encourages efficient use of the low-dimensional embedding space. We also chose to use a denoising reconstruction objective, training the VAE to reconstruct a denoised version of the input image, to encourage the VAE to learn the low-dimensional structure of the data.

The encoder had the same architecture as used in the supervised neural network model for experiments on MNIST and orientation discrimination ("Encoder"), except that instead of a final layer with 100 units, the encoder had two output layers, each of which had only 2 units, to parameterize the means and variances of the latent posterior $q(\mathbf{z}|\mathbf{x})$. The decoder took a sample from this distribution as input, and passed it through 2 FC layers, followed by 3 convolutional layers. The FC layers had 128 and 256 units. The convolutional layers used transposed convolutions to increase the size of the feature map by a factor of 2 at each layer. All convolutional layers used a kernel size of 4. The first 2 convolutional layers had 32 channels, and the final convolutional layer had a single channel. All layers in the decoder used leaky ReLU

nonlinearities with a negative slope of 0.01, except for the output layer, which used a tanh function so as to produce outputs with values in the range $[-1, 1]$ (the same range as the input images). Each VAE was trained for 20 epochs on a two-choice MNIST training set, with a batch size of 32 and a learning rate of $5e-4$, using the standard range of contrast and noise values ("Datasets"). Mean-squared error was used as a reconstruction loss function, and was combined with the KL divergence regularization term. 100 VAEs were trained with different random initializations.

**Ideal observer.** For each trained VAE, we fit $p(\mathbf{z}|\mathbf{x}_{s1})$ and $p(\mathbf{z}|\mathbf{x}_{s2})$, bivariate Gaussian distributions for classes s1 and s2 in the learned two-dimensional latent space. To quantify the variance structure of these distributions, we performed PCA on each distribution separately to identify the major and minor axes, then measured the variance along each of these axes and computed their ratio $\sigma_{target}/\sigma_{nontarget}$. We also computed $\theta_{s1,s2}$, the angular difference between the major axes for the s1 and s2 distributions. Bayes rule was used to compute $p(y=s1|\mathbf{z})$ and $p(y=s2|\mathbf{z})$, the probability that the current stimulus belonged to classes s1 or s2:

$$p(y=s1|\mathbf{z}) = \frac{p(\mathbf{z}|\mathbf{x}_{s1})p(y=s1)}{p(\mathbf{z}|\mathbf{x}_{s1})p(y=s1) + p(\mathbf{z}|\mathbf{x}_{s2})p(y=s2)} \tag{11}$$

$$p(y=s2|\mathbf{z}) = 1 - p(y=s1|\mathbf{z}) \tag{12}$$

A prior of $p(y=s1) = p(y=s2) = 0.5$ was used. Choices were made according to:

$$\hat{y} = \text{argmax}\,(p(y=s1|\mathbf{z}), p(y=s2|\mathbf{z})) \tag{13}$$

Confidence was computed according to:

$$p(\hat{y}=y) = \max(p(y=s1|\mathbf{z}), p(y=s2|\mathbf{z})) \tag{14}$$

For each of the experiments described below, we fit separate latent distributions for the images from the test set. Choices and confidence were computed by using these test distributions to evaluate (i.e., compute a weighted average of) the training distributions for $p(y=s1|\mathbf{z})$, $p(y=s2|\mathbf{z})$, and $p(\hat{y}=y|\mathbf{z})$.

### Experiments and analyses

Unless otherwise noted, experiments were performed over 100 trained networks with different random initializations.

**Meta-d'.** To assess the metacognitive performance of our model, we used meta-d', a recently developed detection-theoretic measure[20]. Just as d' measures the extent to which decisions discriminate between two stimulus classes, meta-d' measures the extent to which confidence ratings discriminate between correct and incorrect trials. Importantly, similar to d', meta-d' is not susceptible to response bias, i.e., the overall rate of high vs. low confidence ratings (except in extreme cases in which the decision-maker responds with either high confidence or low confidence on all trials). For these analyses, we used the python implementation available at: http://www.columbia.edu/~bsm2105/type2sdt/.

**PE bias.** We tested our model for the presence of two related, but distinct, effects, both of which have previously been referred to as the positive evidence bias (PE bias). In one effect (version 1), a classification task is performed with two conditions, one with low contrast/low noise stimuli (low PE condition), and one with high contrast/high noise stimuli (high PE condition), such that decision accuracy is balanced between the two conditions. To test for the this effect, we evaluated networks at two different noise levels, performing a search over a range of contrast levels to identify conditions with balanced accuracy. For MNIST, the noise levels for the low and high PE conditions were $\sigma=1$ and $\sigma=2$; for CIFAR-10, the noise levels were $\sigma=0.1$ and $\sigma=0.2$; and for the orientation discrimination task, the noise levels were $\sigma=0.5$ and $\sigma=1$. For each of the two conditions, we performed a search over 500 contrast levels. For MNIST and the orientation discrimination task, these contrast levels ranged from $\mu=0$ to $\mu=1$. For CIFAR-10, these contrast levels ranged from $\mu=0$ to $\mu=0.2$. For MNIST and CIFAR-10, networks were evaluated on the entire test set for each pair of $\sigma$ and $\mu$ values. For the orientation discrimination task, networks were evaluated on 10,000 trials for each pair of values.

We computed average decision accuracy across the networks trained on each task, for each pair of noise and contrast levels. For the orientation discrimination task, decision accuracy was computed by ignoring the OPT-OUT response, and selecting the argmax of the distribution over LEFT and RIGHT actions (rather than sampling from this distribution probabilistically as in training). For each noise level, we identified the contrast level with average decision accuracy closest to a target performance level. The target performance level was set to the threshold halfway between chance performance and 100% accuracy, since this is generally the most sensitive range for observing psychophysical effects. For MNIST and CIFAR-10, in which chance performance is 10% accuracy, the target performance was 55% accuracy. For the orientation discrimination task, in which chance performance is 50% accuracy, the target performance was 75% accuracy.

This procedure identified higher contrast values for higher noise values, and vice versa, resulting in a balanced signal-to-noise ratio across conditions. The procedure identified contrast values of $\mu=0.27$ (for $\sigma=1$) and $\mu=0.54$ (for $\sigma=2$) for MNIST, contrast values of $\mu=0.08$ (for $\sigma=0.1$) and $\mu=0.16$ (for $\sigma=0.2$) for CIFAR-10, and contrast values of $\mu=0.3$ (for $\sigma=0.5$) and $\mu=0.48$ (for $\sigma=1$) for the orientation discrimination task. We computed the mean and standard error of decision accuracy for these two conditions, and also performed paired $t$-tests. This confirmed that decision accuracy was indeed balanced between the low and high PE conditions.

Finally, for MNIST and CIFAR-10, we compared confidence in the low and high PE conditions, by computing the mean and standard error over all networks trained on each task, and by performing paired $t$-tests. For the orientation discrimination task, we computed the opt-out rate in each condition, where opt-out trials were those on which the OPT-OUT response was the argmax of the distribution over all three actions. We computed the mean and standard error of the opt-out rate over all networks trained on this task, and performed paired $t$-tests.

We also tested our model on a second version of the PE bias (version 2). To test for this effect, we first trained the model on a standard two-choice classification task, in which each trial involved the presentation of either s1 or s2, using the standard range of contrast and noise values (described in 4.2). For the MNIST and CIFAR-10 datasets (usually a 10-choice classification task), each network was trained on a randomly selected pair of classes. After training, we evaluated the model on images containing both classes s1 and s2 superimposed, with different contrast levels. After applying the separate contrast values to the image of each class, the two classes were superimposed according to the following formula:

$$\mathbf{x}_{combined} = \max(\mathbf{x}_{s1}, \mathbf{x}_{s2}) \tag{15}$$

such that the value of each pixel in the combined image was the maximum value of that pixel for the image of each class. We treated the decision output of the model (trained to discriminate between classes s1 and s2) as the model's decision about which class had a higher contrast.

For this version of the effect, the low and high PE conditions were defined by the target contrast, the contrast of the stimulus

corresponding to the correct answer (the higher contrast of the two superimposed stimuli). For all three tasks, the low PE condition had a target contrast of $\mu_{target} = 0.5$ and the high PE condition had a target contrast of $\mu_{target} = 1$. Images were presented at a noise level of $\sigma = 1.5$ for MNIST, $\sigma = 0.6$ for CIFAR-10, and $\sigma = 0.75$ for the orientation discrimination task.

We performed a search over values for the nontarget contrast, the contrast of the stimulus corresponding to the incorrect answer (the lower contrast of the two superimposed stimuli), in order to identify conditions under which accuracy was balanced between the low and high PE conditions. We targeted a performance level of 75% accuracy for all three tasks, halfway between chance performance (50%) and 100% accuracy. For each target contrast, we searched over 500 nontarget contrast values ranging from $\mu_{nontarget} = 0.1$ to $\mu_{nontarget} = 1$ (excluding values that were greater than the target contrast). For MNIST, this resulted in nontarget contrasts of $\mu_{nontarget} = 0.27$ in the low PE condition and $\mu_{nontarget} = 0.7$ in the high PE condition. For CIFAR-10, this resulted in nontarget contrasts of $\mu_{nontarget} = 0.32$ in the low PE condition and $\mu_{nontarget} = 0.67$ in the high PE condition. For the orientation discrimination task, this resulted in nontarget contrasts of $\mu_{nontarget} = 0.18$ in the low PE condition and $\mu_{nontarget} = 0.63$ in the high PE condition. Finally, we computed the mean and standard error of confidence (or the opt-out rate, in the case of the orientation discrimination task) over all networks, and performed paired $t$-tests between the low and high PE conditions.

To test the ideal observer for the PE bias, we applied the same search over stimulus parameters to identify conditions with balanced decision accuracy. Since the ideal observer was only formulated for two-choice tasks, we tested for both versions of the PE bias using the two-choice variant of MNIST (whereas we tested the neural network model for version 1 of the PE bias on the full 10-choice MNIST test set). We targeted a performance level of 75% accuracy. For version 1 of the PE bias, this identified contrast values of $\mu = 0.21$ (for $\sigma = 1$) and $\mu = 0.36$ (for $\sigma = 2$). For version 2 of the PE bias, this identified contrast values of $\mu_{nontarget} = 0.28$ (for $\mu_{target} = 0.5$) and $\mu_{nontarget} = 0.68$ (for $\mu_{target} = 1$).

**Dissociation between type-1 and type-2 sensitivity.** To test for the dissociation between type-1 and type-2 sensitivity identified by Maniscalco et al.[13], we used a two-choice version of the MNIST dataset. For each trained network, two randomly selected digit classes s1 and s2 were used. After training networks on this two-choice task, using images with the standard range of contrast and noise values (described in "Datasets"), networks were evaluated on five conditions. In each of these five conditions, images belonging to class s2 were presented at one of five contrast values $\mu_{i=1}$ through $\mu_{i=5}$. Images from class s1 were always presented at the intermediate contrast $\mu_{i=3}$. The noise level was set to $\sigma = 2$ in all conditions.

Contrast values were fit so as to reproduce the d' values observed in ref. 13, using the procedure described in ref. 37. First, we fit the intermediate contrast $\mu_{i=3}$. To do so, we evaluated all trained networks on 200 contrast levels ranging from $\mu = 0.25$ to $\mu = 0.45$. For each contrast level, we evaluated networks on the entire test set (presenting both s1 and s2 at the same contrast) and computed the average d'. We then used linear interpolation to identify a contrast value corresponding to the target d' for this condition. This resulted in an intermediate contrast of $\mu_{i=3} = 0.36$. We then evaluated the networks again, presenting s1 at this intermediate contrast, and presenting s2 at a range of 1000 contrast levels from $\mu = 0$ to $\mu = 1$. For each contrast, we again evaluated networks on the entire test set, computed the average d', and used linear interpolation to identify contrast values corresponding to the target d' for the other conditions. This resulted in contrast values for those conditions of $\mu_{i=1} = 0.05, \mu_{i=2} = 0.16, \mu_{i=4} = 0.58$, and $\mu_{i=5} = 0.81$. We then evaluated networks on the entire test set for each of these five conditions, recording the trial-by-trial decision and confidence outputs generated by the networks.

Finally, we fit a type-2 noise parameter $\xi$, intended to model the additional accumulation of noise between the time at which decisions and confidence ratings are made. Type-2 noise was incorporated by mapping trial-by-trial confidence ratings into the range $[-\infty, \infty]$ using the logit function, adding Gaussian noise with variance $\xi$, and then mapping back to the range $[0, 1]$ using the sigmoid function. We computed meta-d' and response-specific meta-d' (meta-d' for s1 vs. s2 responses) for all five conditions, performing a grid search across 20 values from $\xi = 0.1$ to $\xi = 2$. We identified the type-2 noise level resulting in the closest fit to the observed meta-d' values, as measured by mean-squared error, resulting in a value of $\xi = 1.2$. We then used the fitted values for $\mu_{i=1..5}$ and $\xi$, and computed the mean and standard error (across all trained networks) of d', meta-d' and response-specific meta-d' in each of the five conditions. We also performed a version of this analysis without the additional noise term $\xi$ (results presented in Supplementary Fig. S2).

We also applied the same procedure to fit stimulus parameters and a type-2 noise parameter for the ideal observer. This resulted in contrast values of $\mu_{i=1} = 0.08, \mu_{i=2} = 0.18, \mu_{i=3} = 0.37, \mu_{i=4} = 0.57$, and $\mu_{i=5} = 0.81$, and a type-2 noise parameter of $\xi = 0.9$.

**Analysis of confidence as a function of sensory evidence space.** To determine whether the neural network's learned confidence strategy was better explained by an RCE heuristic or the latent ideal observer, we employed a generalized version of the task used to test for version 2 of the PE bias. Specifically, after training networks on the two-choice classification variant of MNIST, using images with the standard range of contrast and noise values (described in "Datasets"), we evaluated them with images containing superimposed digits belonging to both classes s1 and s2. For instance, if a network was trained to discriminate the digits 7 vs. 2, we presented that network with images containing overlapping 7's and 2's. The contrast values for each digit class, $\mu_{s1}$ and $\mu_{s2}$, were independently manipulated. We evaluated networks on the entire test set for 100 contrast values between $\mu = 0.1$ and $\mu = 1$, testing each combination of values for $\mu_{s1}$ and $\mu_{s2}$, yielding 10,000 combinations. For evaluation, the noise level was set to a value of $\sigma = 1.5$. We treated the decision output of the network (trained to discriminate between classes s1 and s2) as the network's decision about which digit class had a higher contrast. We then computed, for each combination of contrast values $\mu_{s1}$ and $\mu_{s2}$, the average decision accuracy and confidence over all trained networks. We also performed the same evaluation for the latent ideal observer model, using the test distributions for each combination of $\mu_{s1}$ and $\mu_{s2}$ to evaluate the ideal observer fit to the training distributions.

We used linear regression models to formally compare the neural network's pattern of confidence behavior to the predictions of the BE (BE $= \mu_{s2} - \mu_{s1}$), RCE (RCE $= \mu_{s1}$ if $\hat{y} = s1, \mu_{s2}$ if $\hat{y} = s2$), and ideal observer models. These regressions were fit for all trained networks, and we computed the average predictions for each regression across all combinations of $\mu_{s1}$ and $\mu_{s2}$. To assess the fit of each regression, we computed the average and standard error of $R^2$ across all trained networks, and we compared the fit of different models by performing paired $t$-tests on their $R^2$ values. We also estimated the noise ceiling of this analysis by fitting regression models for each network using the average pattern across all trained networks as a predictor.

Because the BE and RCE rules compute confidence as a linear function of the sensory evidence (whereas the ideal observer computes confidence according to a nonlinear function), we reasoned that it may be useful to apply a logit function to transform the networks' confidence outputs from the range $[0, 1]$ to $[-\infty, \infty]$. We compared two versions of each regression model, one with and one without this logit transformation, and selected the version that performed best. Only the BE model was helped by this transformation, so this is the only model that uses it in the results that we present.

**Alternative training and test regimes.** We compared the standard training regime (contrast sampled from the range $[\mu = 0.1, \mu = 1]$, noise sampled from the range $[\sigma = 1, \sigma = 2]$) to three alternative regimes. In the fixed $\mu$ regime, contrast was set to a fixed value of $\mu = 0.5$, while noise was sampled from the standard range $[\sigma = 1, \sigma = 2]$. In the fixed $\sigma$ regime, contrast was sampled from the standard range $[\mu = 0.1, \mu = 1]$, while noise was set to a fixed value of $\sigma = 1.5$. In the fixed $\mu/\sigma$ regime, contrast was sampled from the standard range $[\mu = 0.1, \mu = 1]$, and noise was set to a fixed ratio of the contrast level $\sigma = 3.75\mu$.

To test for the impact of these regimes on the presence of the PE bias, we trained networks on the 10-choice version of MNIST, sampling contrast and noise values according to one of the four regimes described above, and then followed the procedure described in "PE bias" to test for version 1 of the PE bias.

To evaluate the metacognitive performance of networks trained on these regimes, when tested on each of the regimes, we trained networks on the two-choice version of MNIST, sampling contrast and noise values according to one of the four regimes described above, and then evaluated networks on the entire test set using contrast and noise values sampled from each of the four regimes. For this experiment, we trained 200 networks on each regime, since statistically significant differences between some of the conditions could not be established with only 100 networks per regime. For each combination of training and test regime, we computed the average and standard error of meta-d' over all networks. For each test regime, we performed two-sample $t$-tests for meta-d' between networks trained on that regime vs. networks trained on each of the other regimes.

**Analysis of learned representations.** To better understand the representations learned by the model, we applied principal component analysis (PCA) to the representations in the network's penultimate layer (the output of the encoder $f$). We evaluated networks trained on the two-choice version of MNIST. Both training and evaluation used the standard range of contrast and noise values (described in "Datasets"). To assess the dimensionality of the learned representations, we computed the average and standard error of the variance explained by each principal component (PC) across all trained networks. This revealed that the variance was almost entirely explained (>97%) by the top two PCs alone. We computed density estimates, using a Gaussian kernel, for stimulus classes s1 and s2 along PC1, and density estimates for correct and incorrect trials along PC2. We also performed regression analyses to determine to what extent PCs 1 and 2 predicted DVs of interest. We performed two logistic regressions: (1) PC 1 as predictor and stimulus class s1 vs. s2 as DV, and (2) PC2 as predictor and correct vs. incorrect as DV. We performed three linear regressions: (1) |PC1| as predictor and PC2 as DV, (2) PC1 as predictor, and a logit transformation of the network's decision output $\hat{y}$ as DV, and (3) PC2 as predictor and a logit transformation of the network's confidence output as DV. For each regression, we computed the average and standard error for $R^2$ across all trained networks.

We also performed an analysis to determine whether variables other than the network's confidence output were subject to version 1 of the PE bias. We applied the analysis described in "PE bias" to three other DVs: (1) |PC1|, (2) PC2, and (3) a rectified version of the network's decision output, which was computed by taking the network's continuous decision output (a value in the range $[0, 1]$ representing the predicted probability that the input image belonged to class s2), and applying a rectification at 0.5, resulting in a value representing the predicted probability that the network assigned to its decision. We also performed an analysis to better understand the representations along PC1 for the low and high PE conditions. For each condition, we computed the difference in the mean along PC1 for s1 and s2 trials, and the average variance along PC1. We also computed density estimates for s1 and s2 trials, for the low and high PE conditions, along PC1. Finally, we applied the analysis described in "Analysis of confidence as a function

of sensory evidence space", presenting images of two superimposed digits, and computing, for each combination of contrast levels, the average value for |PC1|, PC2, and the rectified decision output.

**Single unit analysis.** We also analyzed the model's learned representations at the single unit level. This analysis was focusd on the version of the model trained on the RL orientation discrimination task. We chose this version of the model since the task mirrored a task used in previous studies of decision confidence at the single neuron level[3]. We first performed the population-level analyses described in the previous section, confirming that the same two-dimensional geometry was present for this version of the model (Supplementary Fig. S11). Then, we performed linear regressions to quantify the extent to which each unit in the penultimate layer was predictive of either decisions or confidence. The output of the network for this version of the model consisted of a three-dimensional categorical probability distribution, in which the first two dimensions corresponded to the two choices (s1 vs. s2), and the third dimension corresponded to the opt-out response. In the first regression, the DV was a logit transformation of the output unit corresponding to choice s2. In the second regression, the DV was a logit transformation of the output unit corresponding to the opt-out response. These regressions provided two metrics for each neuron: $R^2_{decision}$ and $R^2_{opt-out}$. We classified neurons as decision neurons if $R^2_{decision} > R^2_{opt-out}$, and classified them as confidence neurons if $R^2_{opt-out} > R^2_{decision}$.

To determine whether decision neurons implicitly encoded confidence, we performed an analysis reported by Kiani & Shadlen[3]. This analysis utilizes a measure of normalized activity, defined as:

$$\mathbf{x}_{i,j}^{norm} = \frac{\mathbf{x}_{i,j} - \min_{i=1..N}(\mathbf{x}_{i,j})}{\sigma_{i=1..N, j=1..M}(\mathbf{x}_{i,j})} \quad (16)$$

where $\mathbf{x}_{i,j}$ is the activation of the $j$th neuron (out of $M$ neurons in a given layer) on the $i$th trial (out of $N$ trials), $\min_{i=1..N}()$ computes the minimum activation for $j$th neuron over all $N$ trials, and $\sigma_{i=1..N, j=1..M}()$ computes the variance for all $M$ neurons over all $N$ trials. We computed this measure for all decision neurons in the penultimate layer of the model, for all trials.

We then determined the preferred stimulus class ($T_{in}$) for each decision neuron based on the slope of the decision regression model (positive slope indicates $T_{in} = s2$, negative slope indicates $T_{in} = s1$). Finally, we computed the average normalized activity in each of three conditions, based on whether the model chose the neuron's preferred stimulus ($T_{in}$), the neuron's non-preferred stimulus ($T_{opp}$), or the opt-out response (also called sure target, $T_S$). We performed paired $t$-tests (across all decision neurons) comparing the average normalized activation for $T_{in}$ vs. $T_S$, and for $T_S$ vs. $T_{opp}$.

We also evaluated how the representations at the single neuron level mapped onto the representations at the population level. To do so, we computed the angle of projection for each neuron onto the top 2 PCs, and compared this with $\Delta R^2 = R^2_{decision} - R^2_{opt-out}$, a measure of the extent to which each neuron was more predictive of either decisions or confidence.

**Decoding analyses.** We applied the decoding analysis devised by Peters et al.[15] to our model in the following manner. First, we trained networks on the two-choice version of MNIST, using images with the standard range of contrast and noise values (described in "Datasets"). We then evaluated networks on the entire test set for each of 500 contrast levels between $\mu = 0.1$ and $\mu = 1$, with a noise level of $\sigma = 2$, and selected the contrast level that resulted in performance closest to a target accuracy of 75%. This procedure identified a contrast level of $\mu = 0.33$. We used images with that contrast level and a noise level of $\sigma = 2$ for training decoders.

For each trained network, we trained a decoder to predict the stimulus class of an image given the resulting state of the network. The input to the decoder was formed by concatenating the activation states for all layers in the network (using flattened versions of the convolutional layers), yielding inputs of size 11,236. The decoder was itself a single neural network layer with a sigmoid nonlinearity, and no bias term. The decoder was trained using a cross entropy loss to predict the class $y$ of an image $\mathbf{x}$ passed to the network. This training was performed for 5 epochs with the same training set used to train the network, using the ADAM optimizer with a learning rate of $5e-4$ and a batch size of 32.

We then simulated the ROC analyses from ref. 15. These analyses depend on having, for any given input image $\mathbf{x}$, an independent estimate of the neural evidence in favor of each stimulus class s1 and s2 (i.e., an estimate of the evidence in favor of s1 independent of the evidence in favor of s2, and vice versa). The following formulation was used to obtain these estimates:

$$e_{s1} = \frac{|\mathbf{w}_{\mathbf{w}<0}|\tilde{\mathbf{x}}_{\mathbf{w}<0}}{N_{\mathbf{w}<0}} \qquad (17)$$

$$e_{s2} = \frac{\mathbf{w}_{\mathbf{w}>0}\tilde{\mathbf{x}}_{\mathbf{w}>0}}{N_{\mathbf{w}>0}} \qquad (18)$$

where $\tilde{\mathbf{x}}$ is the concatenated state of all layers in the network following presentation of the image $\mathbf{x}$, $\mathbf{w}$ are the weights of the trained decoder, $\mathbf{w} < 0$ are indices for weights that are $<0$, $\mathbf{w} > 0$ are indices for weights that are $>0$, $N_{\mathbf{w}<0}$ is the number of dimensions with decoder weights $<0$, and $N_{\mathbf{w}>0}$ is the number of dimensions with decoder weights $>0$. Only images from the test set were used for these analyses.

These values were then used as inputs to four ROC analyses. For each analysis, an independent variable (IV) was generated by applying a decision rule to the neural evidence estimates, and both ROC curves and choice probability (area under the ROC curve) were computed to determine to what extent that IV could predict a binary dependent variable (DV). The first analysis determined to what extent the BE rule could predict the network's decisions. The IV was the balance-of-evidence, $e_{s2} - e_{s1}$, and the DV was the network's decision output (target = s2 decision). The second analysis determined to what extent the RCE rule could predict the network's decision outputs. For that analysis, two ROC curves were computed and averaged together: (1) using $e_{s1}$ as an IV, and s1 decisions as targets, and (2) using $e_{s2}$ as an IV, and s2 decisions as targets. The third analysis determined to what extent the BE rule could predict the network's confidence outputs. The IV was the difference between the evidence in favor of the decision made and the evidence against the decision made, i.e., $e_{s1} - e_{s2}$ if an s1 decision was made and $e_{s2} - e_{s1}$ if an s2 decision was made, and the DV was the network's confidence outputs (target = confidence > 0.75). The fourth analysis determined to what extent the RCE rule could predict the network's confidence outputs. The IV was the response-congruent-evidence, i.e., $e_{s1}$ if an s1 decision was made and $e_{s2}$ if an s2 decision was made, and the DV was the the network's confidence outputs (target = confidence > 0.75). For each analysis, the average and standard error of the ROC curves and choice probability were computed across all trained networks. Additionally, paired $t$-tests were performed to determine whether there was a statistically significant difference between decision rules (BE vs. RCE) for either decisions or confidence, and whether there was a statistically significant interaction between decision rule and DV (decisions vs. confidence). Finally, we also carried out an alternative version of this analysis using only the activities from the penultimate layer as input to the decoder (Supplementary Fig. S14).

**Simulating TMS.** To simulate TMS, we added Gaussian noise with variance $\xi$ to the activations in a specific layer in the network. To simulate TMS to V1, we added this noise to the first layer of the network. After training networks on the two-choice variant of MNIST, with the standard range of contrast and noise values (described in "Datasets"), we evaluated these networks at 5 contrast levels from $\mu = 0.1$ to $\mu = 1$, with a noise level of $\sigma = 2$. We tested 5 levels of simulated TMS intensity from $\xi = 1$ to $\xi = 5$, and computed average confidence and d′ in each condition across all 100 trained networks. To simulate TMS to dlPFC, we added noise to the penultimate layer of the network. We evaluated networks with a contrast level of $\mu = 0.55$ and a noise level of $\sigma = 2$. We tested the same levels of simulated TMS intensity, and computed average d′ and meta-d′ across all trained networks.

**Blindsight.** To test for the presence of blindsight-like effects, we used networks trained on the two-choice version of MNIST, using contrast values sampled from the range $[\mu = 0.1, \mu = 1]$ and noise values sampled from the range $[\sigma = 3, \sigma = 4]$. After training, we simulated a lesion to the first layer of the network by multiplying the activities in that layer by a scaling factor of 0.01. We then evaluated the network on the entire test set at five contrast levels from $\mu = 0.5$ to $\mu = 0.9$, with a noise level of $\sigma = 4$. For each contrast value, we computed the mean and standard error for both d′ and meta-d′ across all trained networks. We also computed density estimates, using a Gaussian kernel, for the network's confidence outputs for correct vs. incorrect trials. These measures were compared to control networks without a lesion.

### Reporting summary
Further information on research design is available in the Nature Portfolio Reporting Summary linked to this article.

## Data availability
Source data are provided with this paper.

## Code availability
All code is available at: https://github.com/taylorwwebb/performance_optimized_NN_confidence.

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

## Acknowledgements

We would like to thank Matan Mazor, Brian Maniscalco, Dobromir Rahnev, Megan Peters, Matthias Michel, and Hongjing Lu for helpful feedback and discussions. This research was supported by the award of NIH postdoctoral fellowship F32MH117972-01A1 to T.W.W.

## Author contributions

T.W.W., K.M., and H.L. conceived project and planned experiments. T.W.W., T.Y.S., and S.R. implemented experiments. T.W.W. analyzed results. T.W.W., K.M., and H.L. drafted manuscript.

## Competing interests

The authors declare no competing interests.
