## [Peer Review File · Nature Communications]

REVIEWER COMMENTS

Reviewer #1 (Remarks to the Author: Overall significance):

This paper proposes to use an artificial neural network (ANN) to understand how to investigate how sensory evidence is used to make a decision and to estimate the confidence in this decision. By training an ANN to classify images and assess the accuracy of its decisions (confidence) they can compare the behavior of the network to patterns observed in confidence reports from human subjects and investigate the network's "confidence reports" under several scenarios.

The approach of using ANNs in this way has gained traction in neuroscience, for example to explore the evolution of representations along the visual cortex hierarchy. The approach of using an ANN to extract the decisions variables underlying choices and confidence is interesting and a logical next step, although not entirely novel for decision confidence (see "Rats use memory confidence to guide decisions", Joo et al., *Current Biology*, 2021, which should be cited).

They show that the network trained to estimate the accuracy of its decision exhibits patterns of confidence reports similar to those of human subjects. They also show that network activity shares some similarities with fMRI activity in a similar task. They do not however directly compare the representations in single units/areas as was done with the visual system. The structure of confidence representations is not as well understood as the structure of representations in the visual hierarchy, but we do have some ideas about single neuron representations in several areas (see for example work from the Shadlen/Kiani and Kepecs groups) and the network could have been used to see if similar representations arise.

An important conclusion is that despite the complexity of the visual stimuli, in the last layer of the ANN, the decision process can be mapped on a 2-dimensional decision variable. This is encouraging for neuroscientists studying decision processes and suggests that the noise across images can be mapped onto a low-dimension and potentially allow experimenters to sample the variations across these dimensions.

Another key conclusion is that seemingly surprising dissociations between choice accuracy and confidence in human subjects also arise in the network. They show that in the network there is a similar relationship between d' (a measure of perceptual accuracy) and meta- d' (metacognitive accuracy) than in human subjects. They also show that these changes in meta- d' are dependent on the distribution of the training data. Together these results are interesting they show that human-like reports do seem to arise in systems optimized to predict their accuracy, but the analysis could go further.

The conclusion they make are observational, but one of the advantages of using a network is the ability to obtain a more mechanistic understanding. A major concern is whether the observed effects would be expected from a Bayesian ideal observer who has only access to its percepts and outcomes given the training distribution. The results from the network make me believe so but the authors do not attempt to show this. The authors use two models (balance of evidence and response congruent evidence) that are symmetric along different dimensions but the asymmetry in the stimulus statistics implies that the Bayesian posterior will not be symmetric. It is therefore difficult to see if these dissociations are not purely the result of a Bayesian posterior probability of being correct that is computed (with some approximation) by both human subjects and the network.

Reviewer #1 (Remarks to the Author: Impact):

This paper is an interesting addition to the body of work from the authors but to have a broader appeal in the neuroscience and psychology communities they should use the ANN approach to describe and further understand results from other groups studying decision confidence and its neural substrates.

Reviewer #1 (Remarks to the Author: Strength of the claims):

Major comments

1. The main concern with the approach presented is that although the network does reproduce a number of “puzzling dissociations between decision and confidence”, it remains unclear if these would not also be obtained with an ideal Bayesian observer computing the posterior probability of being correct given the available subjective evidence, the choice and the training data. The asymmetric mean and standard deviation of the noise across conditions introduce asymmetries in the posterior. Without this result, it is hard to assess whether these dissociations are truly puzzling or simply the result of Bayesian inference with asymmetric information across options. For example, the simulations in “A model of subjective report and objective discrimination as categorical decisions in a vast representational space”, King and Dehaene, *Philos. Trans. R. Soc. B*, 2014, (which should be cited) provide some examples of such asymmetries.

The authors should use a Bayesian decision model and find the appropriate task description in terms of stimulus difficulty distribution and noise distributions (see for example, King and Dehaene, *Philos. Trans. R. Soc. B*, 2014, Sanders et al. *Neuron*, 2016, Hangya et al., *Neural Computation*, 2016, Adler and Ma, *Neural Computation*, 2018 or Rausch and Zehetleitner, *Plos Comp Bio*, 2019). This analysis could uncover whether the observed asymmetries could be simply related to the asymmetry in the quality of the evidence across task conditions. Furthermore, in the tasks that manipulate contrast, we would expect the network to be able to learn the contrast level independently of the choice and therefore use this information as part of the posterior estimate as described in Rausch and Zehetleitner, 2019.

2. Related to this first point, the authors compare the responses to two heuristic rules but due to the asymmetries in the tasks they find that a combination of both rules capture better the confidence reports. As a first step (instead of the full Bayesian model), is there a relationship between the deviation from fully correlated d' -meta d' and the relative weights of the two rules in a combined model?

3. The authors should present all confidence reports using the histograms of confidence distributions for correct/error rather than the error bars. Mean confidence is difficult to interpret.

Minor comments

4. The neural network model should be described in the first subsection of the results section. Currently, the description is entirely in the legend of Figure 1 and methods. A paragraph or two with a slightly extended version of the legend would be helpful.

5. The comparison with lesions and stimulations should perhaps be placed nearby in the text to give a more coherent view on the experimental data that could be studied by the ANN approach.

6. The authors do not really distinguish between experiments involving accumulation of evidence and those involving “static” stimuli in their discussions. In accumulation tasks, the amount of “post-decisional” evidence can vary according to experimental design and a direct comparison can be difficult. The authors should highlight those points.

7. Figures 7b, S5c and S7b,c. I would change the color scheme for s1/s2 so it is different than for the panels showing correct/incorrect distributions in the rest of the paper. These panels show different information, and a similar color scheme might confuse the reader.

Reviewer #1 (Remarks to the Author: Reproducibility):

This work is based on model simulations, and I commend the authors for providing their code with detailed instructions in a GitHub repository, which should allow anyone to reproduce their results. In addition, the training procedure are well described in the methods.

They should however:

1. Include the full URLs in the methods section rather than links through Hyperlinks.
2. Add some information in the methods about the hardware used for the simulations.

Reviewer #2 (Remarks to the Author: Overall significance):

I’m glad of the opportunity to review this paper which I have seen in multiple iterations both as a preprint and conference presentation at CogSci. Each time I see these results I continue to think what an impressive and counterintuitive piece of work it is. I am often skeptical of the value of “black box” machine learning models of cognition, but this one is different in that it targets a common assumption in the field (and one I believe was originally held by the senior author of this paper!) – that a distinct mechanism or representation may need to be posited to explain biases in confidence and metacognition. The use of a neural network to show that these biases arise due to constraints imposed by the (higher dimensional) training data is therefore very informative indeed and a beautiful example of how these methods can be brought to bear on foundational questions in psychology and systems neuroscience.

My expertise is on the confidence / metacognition side and not neural network modelling, so I will not attempt to critique or evaluate the methods. I have a few comments, mostly minor, that I will list in order of occurrence in the paper.

Reviewer #2 (Remarks to the Author: Impact):

Yes - it will have a substantial impact on thinking in the fields of perceptual decision-making, confidence and metacognition. I think it deserves a wide readership in a general audience neuroscience journal.

Reviewer #2 (Remarks to the Author: Strength of the claims):

1) At a few points in the introduction the RCE rule is introduced as all or none – “...only considers” the response-congruent evidence, or “confidence is based only on the evidence in favour...”. This strikes me as an overly strong characterisation of the empirical literature, where the impact of response-incongruent evidence on confidence is often downweighted but not abolished entirely (and in fact, this is what the neural network shows too, eg Figure 5)

2) It struck me that, compared to state-of-the-art image classification, the performance in the MNIST and CIFAR databases was notably low (eg 55% correct in Figure 2d, where chance is presumably 10%). What is the reason for this? Is it artificially constrained by the noise in the image? Or due to the relatively simple neural network architecture for image classification? Some comment about how or whether performance could be boosted, and whether this would affect the confidence patterns would be useful (eg I am guessing that for near-ceiling performance, the confidence biases might diminish as most trials are predicted to be correct).

3) “Blindsight patients have very low confidence in their visual discriminations”. Is this true? My understanding from the cited Persaud et al. study was that overall confidence (the average proportion of high gambles) was similar in the blind and sighted fields, but that metacognitive sensitivity was impaired.

4) Figure 3 – why is meta- d' systematically negative in the lesion condition?

5) The analysis in Figure 4 of the drivers of the bias is informative. It struck me that the explanation being put forward is compatible with classical cue-based inferential models of confidence in the memory literature. In other words, the neural network appears to be capitalising on any feature of the input that usually tracks with confidence – similar to how fluency becomes a useful predictor of memory confidence. Is this analogy correct / useful? Ironically, if yes, perhaps the more recent metaperception literature was misled by a focus on Bayesian latent states that are directly coupled to features of the stimulus, rather than cues more broadly construed.

6) Khalvati et al. (ref. 37) also propose that a common decision variable can underpin dissociations between performance and confidence, including the positive evidence bias. However, they appeal to a somewhat different mechanism – Bayesian inference with partially observable state information. Can this difference in perspective be reconciled or commented on?

We would like to thank the reviewers for their many insightful comments and suggestions. We have now revised the manuscript to address these issues. In particular, in response to a suggestion from reviewer 1, we have implemented a Bayesian ideal observer that replicates the confidence biases we investigate and very closely matches the pattern of our neural network model. We have reframed the paper accordingly, resulting in a much more decisive account of these phenomena. Below we include a point-by-point reply to the issues raised by the reviewers, along with a description of the corresponding revisions made to the manuscript. The reviewers' comments are presented in blue, and revisions are presented in red.

Reply to reviewer 1

The main concern with the approach presented is that although the network does reproduce a number of “puzzling dissociations between decision and confidence”, it remains unclear if these would not also be the obtained with an ideal Bayesian observer computing the posterior probability of being correct given the available subjective evidence, the choice and the training data. The asymmetric mean and standard deviation of the noise across conditions introduce asymmetries in the posterior. Without this result, it is hard to assess whether these dissociations are truly puzzling or simply the result of Bayesian inference with asymmetric information across options.

We would like to thank the reviewer for urging us to more directly address the question of optimality. We have now implemented a Bayesian ideal observer model that captures the same behavioral dissociations as the neural network model. The key challenge in doing so was to characterize the stimulus distributions to which the ideal observer is applied, which can have a dramatic effect on whether the model displays these dissociations. To address that issue, we used an unsupervised deep learning method (variational autoencoders) to extract a low-dimensional representation of the neural network model's training data, to which we could then directly apply an ideal observer. In addition to robustly capturing the behavioral dissociations between decisions and confidence, the ideal observer also very closely matched the network's confidence pattern as a function of the sensory evidence space. These results point to a rational explanation of these dissociations in terms of the statistics of sensory experience. We have now reframed the paper to emphasize this conclusion. We enumerate the revisions related to this analysis below:

- 1) We have changed the title of the paper to '*Natural statistics support a rational account of confidence biases*'.
- 2) We have modified a paragraph in the introduction, and added a figure to compare the Bayesian ideal observer model with the BE and RCE decision rules:

In particular, it has been shown that, given alternative assumptions about the variance structure governing stimulus distributions, the optimal approach to estimating confidence entails a more complex function that differs from both the BE and RCE rules (Figure 1c) [19], with some evidence that human decision confidence follows this pattern [18]. However, it has yet to be shown whether this alternative model can account for the previously observed

dissociations between decisions and confidence. More importantly, this alternative model calls attention to the fact that questions about optimality must be framed in relation to stimulus distribution structure, which has typically been treated as a modeling assumption in previous work.

Figure 1: Detection-theoretic formalization of confidence in two-choice tasks. (a) Stimuli are modeled as samples from two-dimensional Gaussian distributions (with means μ_{s1} and μ_{s2} , and variance σ), schematized as circles labeled 's1' and 's2', where each dimension represents the evidence in favor of one stimulus category. Given these assumptions, the optimal procedure for estimating confidence is a 'balance-of-evidence' (BE) rule, based on the difference between the evidence in favor of s1 and s2. **(b)** Many results are well modeled by an alternative 'response-congruent-evidence' (RCE) heuristic, according to which, after making a decision, confidence is based entirely on the evidence in favor of the chosen stimulus category, ignoring the evidence in favor of the alternative choice. **(c)** Bayesian ideal observer with alternative variance assumptions. When stimulus distributions are characterized by greater variance in the dimension in favor of the correct answer (σ_{target}) than the dimension in favor of the incorrect answer ($\sigma_{\text{nontarget}}$), as proposed in [18] and [19], the optimal procedure for estimating confidence involves a more complex function.

[18] L. Aitchison, D. Bang, B. Bahrami, and P. E. Latham, "Doubly bayesian analysis of confidence in perceptual decision-making," PLoS computational biology, vol. 11, no. 10, e1004519, 2015.

[19] K. Miyoshi and H. Lau, "A decision-congruent heuristic gives superior metacognitive sensitivity under realistic variance assumptions.," Psychological Review, vol. 127, no. 5, pp. 655–671, 2020.

3) We have modified the paragraph in the introduction describing our contributions:

In this work, we developed a model of decision confidence that operates directly on naturalistic, high-dimensional inputs, avoiding the need for these simplifying assumptions. To do so, we first developed a performance-optimized neural network model trained both to make decisions from high-dimensional inputs, and to estimate confidence by predicting the

probability those decisions will be correct. Surprisingly, a number of seemingly suboptimal features of confidence naturally emerged from the model, including the 'positive evidence' bias. We then used unsupervised deep learning methods to extract a low-dimensional representation of the model's training data. We found that the training data distribution displayed key properties that undermined the presumed optimality of the BE model, and that an ideal observer applied to this distribution replicated the observed dissociations, thus yielding a rational account of these dissociations. Consistent with this, we found that altering the distribution of the training data altered the resulting biases in predictable ways, and that the model employed a common internal decision variable for both decisions and confidence, despite the observed behavioral dissociations. Finally, we found that the model also accounts for a range of neural dissociations between decisions and confidence, including some features akin to 'blindsight' resulting from lesions to the primary visual cortex. These results provide a novel perspective on the computational and neural basis of decision confidence, and suggest new avenues for future investigation.

- 4) We now refer to the low-dimensional latent representations extracted by the autoencoder as \mathbf{z} . Therefore, to avoid confusion, we have removed references to the penultimate layer in the supervised neural network as \mathbf{z} , and revised the figure depicting the supervised neural network accordingly.
- 5) We have included simulations testing the model for an alternative version of the positive evidence bias, because we found that the ideal observer model had different implications for these two versions of the bias. Specifically, for the version of the PE bias that we originally investigated, involving two conditions with differing levels of signal and noise, we found that the ideal observer model captured this effect regardless of the shape of the stimulus distributions. However, for an alternative version of the bias, involving superimposed stimuli belonging to the two stimulus classes, the ideal observer only showed this bias when the stimulus distributions were characterized by the key feature of unequal variance in the target vs. nontarget dimensions. We found that the supervised neural network model showed both versions of this bias:

Confidence is characterized by a positive evidence (PE) bias [11,12,14,16,17], as revealed by two related, but distinct, manipulations. In one version of this effect, participants are presented with two conditions, one with low signal and low noise ('low PE' condition, Figure 3a), and the other with high signal and high noise ('high PE' condition). In the other version of this effect, participants are presented with a two-choice task involving stimuli that contain some evidence in favor of both choices, and have to decide which choice has more evidence in favor of it. For example, in the conditions depicted in Figure 3c, the task is to decide which of two superimposed digits ('4' and '6' in this example) has a higher contrast. The 'high PE' condition has both higher positive evidence (evidence in favor of the correct answer) and higher negative evidence (evidence in favor of the incorrect answer) than the 'low PE' condition. In both versions of the effect, the PE bias manifests as higher confidence in the high vs. low PE conditions, despite the fact that signal-to-noise ratio, and therefore decision accuracy, is balanced across these conditions. This bias is considered a key piece of evidence against the BE model (Figure 1a), and in favor of the RCE model (Figure 1b),

since the BE model gives equal weight to the evidence both for and against a decision, whereas the RCE model considers only the evidence in favor of a decision.

Figures 3b and 3d show that both versions of the PE bias naturally emerged in our model across a range of conditions. For both the MNIST and CIFAR-10 datasets, confidence was higher in the high vs. low PE conditions, despite balanced accuracy, as previously observed in studies of human decision confidence [11,12,17]. The presence of this bias therefore did not depend on the specific dataset used, or the architectural details of the model, since experiments on CIFAR-10 used a more complex ResNet architecture for the encoder. In the orientation discrimination RL task, the opt-out rate was lower in the high vs. low PE conditions, as previously observed in studies using animal models [14,16]. The presence of this bias therefore did not depend on the use of supervised learning to train the confidence layer, but also emerged when using a more realistic training signal (reward).

Figure 3: Behavioral dissociations between decisions and confidence. (a) Human and animal decision confidence displays a positive evidence (PE) bias: higher confidence (or lower opt-out rate) in the high vs. low PE conditions despite balanced signal-to-noise ratio and balanced decision accuracy. (b) The PE bias naturally emerges in performance-optimized neural networks across multiple datasets, architectures, and learning paradigms. (c) An alternative test for the PE bias, involving superimposed stimuli presented at different contrast levels, where the task is to indicate which stimulus is presented at a higher contrast. In the 'high positive evidence' condition, there is both higher positive evidence (evidence in favor of the correct answer, '4' in this case), and higher negative evidence (evidence in favor of the incorrect answer, '6' in this case), than in the 'low positive evidence' condition. Visual noise was also included in images, but is omitted here for clarity of visualization. (d) The model also shows this alternative formulation of the PE bias.

6) We have included a new section describing the ideal observer model, and showing that it accounts for the key behavioral dissociations between decisions and confidence:

2.2 Latent ideal observer

The previous results show that our model captures a number of established behavioral dissociations between confidence and decision accuracy. How can these dissociations be explained? One possibility is that, despite being extensively optimized to estimate confidence by predicting its own probability of being correct, the model nevertheless converged on a suboptimal heuristic strategy. An alternative possibility is that these effects reflect a strategy that is optimal given the actual distribution of the data for which the model was optimized, which may violate the assumptions underlying the presumed optimality of the BE rule. We found strong evidence to support this latter interpretation.

To answer this question, we first sought to quantitatively characterize the distribution of the model's training data. Because it is not tractable to perform ideal observer analysis directly on the model's high-dimensional inputs, we instead used unsupervised deep learning techniques to extract the low-dimensional, latent space underlying those inputs. Specifically, we used a denoising variational autoencoder (VAE; Figure 4a), which was trained to map a high-dimensional input \mathbf{x} to a low-dimensional embedding \mathbf{z} (consisting of just two dimensions), such that a denoised version of \mathbf{x} can be decoded from \mathbf{z} . Figure 4b depicts a summary of the low-dimensional latent distributions extracted by the VAE. These distributions had two important properties. First, these distributions had an elliptical shape, as quantified by the ratio of the variance along the major and minor axes ($\sigma_{\text{target}} / \sigma_{\text{nontarget}} = 2.44 \pm 0.04$ over 100 trained VAEs). Second, the distributions underlying classes s_1 and s_2 fell along non-parallel axes ($\theta_{s_1, s_2} = 51.4^\circ \pm 1.7$). Under these conditions, the optimal approach for estimating confidence follows a more complex function than either the BE or RCE rules, as visualized in Figure 4b.

We then constructed an ideal observer that computed confidence according to $p(\text{correct}|\mathbf{z})$, using the distribution of the training data in the low-dimensional space extracted by the VAE, and we evaluated this function according to the distribution of the test data in this space. This ideal observer model robustly captured both versions of the PE bias (Figures 4c and 4d), as well the dissociation between type-1 and type-2 sensitivity (Figure 4e), thus replicating the same dissociations displayed by our performance-optimized neural network model. A more comprehensive analysis is presented in Supplementary Figures S5 and S6, showing that the emergence of these biases depends on sensory evidence distributions that are *both* asymmetrical ($\sigma_{\text{target}} / \sigma_{\text{nontarget}} > 1$), and non-parallel (as quantified by $0^\circ < \theta_{s_1, s_2} < 180^\circ$).

Importantly, we found that a key driver of this variance structure was the presence of variable contrast in the model's training data. When the training data involved only images presented at a fixed contrast, the distributions extracted by the VAE were characterized by asymmetric variance ($\sigma_{\text{target}} / \sigma_{\text{nontarget}} = 2.57 \pm 0.06$), but with a much smaller angular difference ($\theta_{s_1, s_2} = 10.8^\circ \pm 1.6$, two-sample t-test, standard training regime vs. fixed-contrast regime, $t=17.2$, $p<0.0001$). In line with this observation, we show in Section 2.3 that

manipulating this feature of the training data has a dramatic effect on the biases displayed by the model.

Figure 4: Latent ideal observer accounts for dissociations between decisions and confidence. **(a)** Denoising variational autoencoder (VAE) used to extract low-dimensional latent representation of training data. An image \mathbf{x} was passed through a DNN encoder q (distinct from the encoder f used in the supervised neural network model), which output the parameters (means and variances) of the latent posterior $q(\mathbf{z}|\mathbf{x})$, a two-dimensional Gaussian distribution. This distribution was sampled from, yielding \mathbf{z} , which was then passed through a DNN decoder h , yielding $\tilde{\mathbf{x}}$, a denoised reconstruction of the input \mathbf{x} . The VAE was regularized based on the divergence of the latent posterior from a unit normal prior (with means equal to 0 and variances equal to 1), encouraging efficient low-dimensional encodings of the high-dimensional inputs. **(b)** Latent ideal observer model. After training the VAE, Gaussian distributions were fit to the latent representations resulting from the training images for classes s_1 and s_2 . The distributions were used to construct an ideal observer model that computed confidence according to $p(\text{correct}|\mathbf{z})$, the probability of being correct given the low-dimensional embedding \mathbf{z} . Concentric ellipses represent distributions based on the average parameters of those extracted from 100 trained VAEs. The latent ideal observer accounted for both versions of the PE bias, including **(c)** the version involving manipulation of contrast and noise (Figure 3a), and **(d)** the version involving superimposed stimuli presented at different contrast levels (Figure 3c). **(e)** The latent ideal observer also accounted for the dissociation between type-1 and type-2 sensitivity. Results for panels **(c-e)** reflect an average over 100 ideal observers (each based on distributions extracted by a separate trained VAE) \pm the standard error of the mean; 'ns' indicates $p > 0.05$, '****' indicates $p < 0.0001$.

Figure S5: Dependence of PE bias on sensory evidence variance structure. Analysis of both versions of PE bias as a function of the variance structure of the sensory evidence distributions. Ideal observer was implemented for training distributions that varied in terms of two properties: 1) $\sigma_{target} / \sigma_{nontarget}$, the ratio of the variance in the target vs. nontarget dimensions (intuitively, the extent to which the training distributions had an elliptical shape), and 2) $\theta_{s1, s2}$, the angular difference between the major axes of the distributions for stimulus classes $s1$ and $s2$. Ideal observer was evaluated on test distributions with equal variance in the target and nontarget dimensions (i.e., circular distributions), with a signal-to-noise ratio corresponding to an accuracy of 75% for all conditions. Colored lines correspond to distributions that varied in both their target mean μ (the mean in the dimension corresponding the correct answer) and their variance σ . Thus, version 1 of the PE bias corresponds to an increase in confidence between the light blue and dark blue lines. This version of the PE bias emerged under all variance conditions. The X axes represent the value of a constant added to both dimensions. Thus, version 2 of the PE bias corresponds to an increase in confidence as a function of the X axis. **(a)** This version of the PE bias only emerged when sensory evidence distributions were asymmetric ($\sigma_{target} / \sigma_{nontarget} > 1$) and non-parallel ($0^\circ < \theta_{s1, s2} < 180^\circ$). When sensory evidence distributions had **(b)** equal variance in both dimensions ($\sigma_{target} / \sigma_{nontarget} = 1$), or had major axes with an angular difference of

either (c) $\theta_{s1,s2} = 0^\circ$ or (d) $\theta_{s1,s2} = 180^\circ$, the ideal observer did not show this version of the PE bias.

Figure S6: Dependence of dissociation between type-1 and type-2 sensitivity on sensory evidence variance structure. (a) Dissociation between type-1 and type-2 sensitivity emerged only when sensory evidence distributions were asymmetric ($\sigma_{\text{target}} / \sigma_{\text{nontarget}} > 1$) and non-parallel ($0^\circ < \theta_{s1,s2} < 180^\circ$). When sensory evidence distributions had (b) equal variance in both dimensions ($\sigma_{\text{target}} / \sigma_{\text{nontarget}} = 1$), or had major axes with an angular difference of either (c) $\theta_{s1,s2} = 0^\circ$ or (d) $\theta_{s1,s2} = 180^\circ$, the ideal observer did not show this dissociation.

- 7) The previous version of the manuscript included a figure showing that the common internal decision variable, used for both decisions and confidence, is subject to the PE bias. We have now moved this figure to the supplementary results, since the same point is made more effectively by the ideal observer model.
- 8) We have revised the first two paragraphs of the discussion to emphasize the rational account provided by the ideal observer model:

The question of whether confidence judgments reflect an optimal prediction of the probability of being correct has been hotly debated in recent years [3-9,11-19,31,32]. Those debates have centered around models that rely on strong assumptions about the representations over which confidence is computed. Previous work using artificial neural networks to model decision confidence has also generally relied on such representational assumptions [33-36]. We found that a relatively simple model, optimized only to predict its own likelihood of being correct, captured many of the biases and dissociations that have driven recent debates. Furthermore, we found that an ideal observer applied to a low-dimensional projection of the model's training data yielded a rational explanation of these biases, and provided a very close fit to the pattern of confidence displayed by the model.

Our findings have an important link to models of decision confidence in which sensory evidence distributions are characterized by asymmetric variance [18,19], according to which the optimal confidence strategy follows a more complex function than a simple BE rule. We found that a low-dimensional projection of our model's training data displayed this key property of asymmetric variance, driven to a significant extent by the presence of variable contrast in the training data. Consistent with this, we found that the PE bias could be eliminated, or even reversed by manipulating this feature of the training data. More broadly, these results suggest that human confidence biases may emerge as a consequence of optimization for a particular statistical regime, in particular one governed by variable signal strength, and it therefore may be possible to reshape them through direct training under specific task conditions. This idea is consistent with the results of Maniscalco et al. [13], who found that the dissociation between type-1 and type-2 sensitivity could be largely eliminated when participants received feedback about the effectiveness of their type-2 responses.

- 9) We have updated the methods and our publicly available GitHub repository to include all additional experiments, including all simulations with the ideal observer (section 4.5), and all experiments testing for the alternative formulation of the PE bias (section 4.6.2).

The reviewer suggested that we cite: “A model of subjective report and objective discrimination as categorical decisions in a vast representational space”, King and Dehaene, *Philos. Trans. R. Soc. B*, 2014

This work is now cited in the introduction:

It has been proposed that this sense of confidence corresponds to an optimal prediction of the probability that a decision will be correct, and that confidence is computed based on the same underlying decision variable as decisions themselves [3-9]. Given certain distributional assumptions, this approach entails the use of a decision variable that is proportional to the 'balance-of-evidence' (BE; Figure 1a), incorporating sensory evidence both for and against a decision [10].

[10] J.-R. King and S. Dehaene, "A model of subjective report and objective discrimination as categorical decisions in a vast representational space," *Philosophical Transactions of the Royal Society B: Biological Sciences*, vol. 369, no. 1641, p. 20 130 204, 2014.

Related to this first point, the authors compare the responses to two heuristic rules but due to the asymmetries in the tasks they find that a combination of both rules capture better the confidence reports. As a first step (instead of the full Bayesian model), is there a relationship between the deviation from fully correlated d' -meta d' and the relative weights of the two rules in a combined model?

We have now replaced the multiple regression model (BE + RCE) with the ideal observer, which is both theoretically better motivated and does a better job of predicting the neural network's confidence outputs:

2.2.1 Comparing the latent ideal observer and RCE models

Given that the observed behavioral dissociations can, in principle, be explained both by the RCE heuristic model and our latent ideal observer model, we next sought to determine which of these models best characterized the behavior of the neural network. To do so, we employed a generalized version of the positive evidence manipulation, evaluating the neural network model across a grid of conditions, each of which was defined by a particular set of contrast levels for stimulus classes s_1 and s_2 (Figure 5a). This allowed for a more comprehensive characterization of the model's behavior as a function of the sensory evidence space.

We found that the model's decision accuracy strongly resembled the BE rule (Figure 5b), whereas confidence displayed a more complex pattern (Figure 5c). To better understand the model's confidence behavior, we formally compared the pattern displayed in Figure 5c to regression models based on the BE and RCE rules, as well as a regression model based on the latent ideal observer. Confidence was better explained by the RCE rule than the BE rule (RCE $R^2=0.82 \pm 0.01$; BE $R^2=0.42 \pm 0.01$; paired t-test, RCE vs. BE, $t=33.37$, $p<0.0001$), but the ideal observer explained confidence better than either of these rules (ideal observer $R^2=0.89 \pm 0.01$, Figure 5d; paired t-test, ideal observer vs. RCE, $t=28.4$, $p<0.0001$), and indeed was very close to the noise ceiling (the ability of the average pattern across networks to predict the behavior of individual networks, $R^2=0.9 \pm 0.01$).

These results give rise to a few questions. First, given that the ideal observer estimates confidence according to an optimal prediction of its own accuracy, what explains the difference between the patterns displayed by accuracy and confidence? This can be explained by the fact that the ideal observer's confidence estimates are based on the probability of being correct given the *training* distribution, which deviates from the distribution of the test conditions evaluated here. In particular, the conditions employed in this evaluation are designed to uniformly sample the sensory evidence space, whereas the training data are not uniformly distributed in this space. Second, why does the confidence pattern displayed

by the ideal observer (Figure 5d) differ from $p(\text{correct}|\mathbf{z})$, the function it uses to compute confidence (Figure 4b)? This is because the test conditions depicted in Figure 5a each contain their own degree of noise, and do not correspond to precise point estimates of the ideal observer's confidence function. Thus, the pattern displayed in Figure 5d reflects essentially a smoothed version of the ideal observer model. Surprisingly, the result bears a strong visual resemblance to the RCE rule, though our quantitative analysis reveals that these two models can be distinguished, and that the ideal observer ultimately provides a better explanation of the confidence strategy learned by the neural network.

Figure 5: Learned confidence strategy best explained by ideal observer. (a) Generalized version of positive evidence manipulation used to comprehensively evaluate both accuracy and confidence as a function of sensory evidence. Model was trained on classification of individual stimuli over the standard range of contrast and noise levels, then tested on images consisting of two superimposed stimuli belonging to classes s_1 and s_2 , with independently varying contrast levels μ_{s1} and μ_{s2} . **(b)** Decision accuracy resembled the BE rule, as expected given uniform sampling of sensory evidence space. **(c)** Confidence displayed a more complex pattern. **(d)** Confidence was best predicted by the latent ideal observer model, which outperformed regression models based on either the RCE or BE rules. Results reflect an average over 100 trained networks.

We have also updated the methods and GitHub repository accordingly.

The reviewer suggested that we cite: “Rats use memory confidence to guide decisions”, Joo et al., Current Biology, 2021

We now cite this study in the discussion, along with a few other previous studies that used neural networks to study decision confidence. The key difference between our work and these previous studies is that the previous studies used assumed low-dimensional representations as inputs to a neural network, whereas we focused on modeling confidence for real images, which turned out to be critical for explaining the biases we investigated:

Previous work using artificial neural networks to model decision confidence has also generally relied on such representational assumptions [33-36].

[33] A. Pasquali, B. Timmermans, and A. Cleeremans, “Know thyself: Metacognitive networks and measures of consciousness,” *Cognition*, vol. 117, no. 2, pp. 182–190, 2010.

[34] H. F. Song, G. R. Yang, and X.-J. Wang, “Reward-based training of recurrent neural networks for cognitive and value-based tasks,” *Elife*, vol. 6, e21492, 2017.

[35] B. Maniscalco et al., “Tuned inhibition in perceptual decision-making circuits can explain seemingly suboptimal confidence behavior,” *PLOS Computational Biology*, vol. 17, no. 3, pp. 1–28, 2021.

[36] H. R. Joo et al., “Rats use memory confidence to guide decisions,” *Current Biology*, vol. 31, no. 20, pp. 4571–4583, 2021.

They do not however directly compare the representations in single units/areas as was done with the visual system. The structure of confidence representations is not as well understood as the structure of representations in the visual hierarchy, but we do have some ideas about single neuron representations in several areas (see for example work from the Shadlen/Kiani and Kepecs groups) and the network could have been used to see if similar representations arise.

We agree that this is an important goal for future work. However, in this work we were focused on the question of dissociations between decisions and confidence. To our knowledge, Odegaard et al. (reference #16 below), is the only study to investigate the correlates of such dissociations at the single neuron level, and the findings of that study were effectively a null result with respect to this question – the neural data recorded from the superior colliculus did not show any signature related to the behavioral dissociation between decisions and confidence. Therefore, it is currently unknown how representations at the single neuron level might support these dissociations. For that reason, we focused on the behavioral and cognitive neuroscience data that specifically relates to our question. We have added a note to the discussion clarifying this issue:

The representational scheme learned by the model, in which both confidence and decisions are represented using a common decision variable, bears a striking resemblance to previous results in decision-making regions such as LIP and PFC [3,40]. In contrast to this common decision variable account, it has recently been proposed that confidence and decisions are supported by distinct populations of neurons with differing degrees of lateral inhibition, accounting for the observed dissociations between decisions and confidence [38]. A key contribution of our model is to show how these dissociations can arise despite the use of a common decision variable. This leads to the novel prediction that the neural decision variable underlying decisions should be subject to the same biases as confidence, even if decisions don't display those biases at a behavioral level. To our knowledge, no previous studies at the single neuron level have successfully identified neural correlates of these dissociations, though one study found that the superior colliculus does not appear to play a role [16]. The identification of these correlates is thus an important goal for future work, as it would enable a direct test of our model.

[16] B. Odegaard, P. Grimaldi, S. H. Cho, M. A. Peters, H. Lau, and M. A. Basso, “Superior colliculus neuronal ensemble activity signals optimal rather than subjective confidence,” *Proceedings of the National Academy of Sciences*, vol. 115, no. 7, E1588–E1597, 2018.

The authors should present all confidence reports using the histograms of confidence distributions for correct/error rather than the error bars. Mean confidence is difficult to interpret.

We agree that it is important to give a richer characterization of the model's confidence reports. We have now included violin plots depicting confidence for correct vs. incorrect trials in the supplementary results (with a pointer to these plots in the corresponding figure legends in the main text), and have added code for these analyses to our GitHub repository:

Figure S1: Confidence for both correct and incorrect trials shows a positive evidence bias. Panels (a) (MNIST), (b) (CIFAR-10), and (c) (RL orientation discrimination task), show confidence for correct and incorrect and trials, for the version of the PE bias involving manipulation of contrast and noise. Panels (d) (MNIST), (e) (CIFAR-10), and (f) (RL orientation discrimination task), show confidence for correct and incorrect and trials, for the version of the PE bias involving superimposed stimuli presented at different contrast levels. In all cases, confidence shows a significant PE bias for both correct and incorrect trials. All results reflect the probability density over 100 trained networks, with mean confidence in each condition represented by circular markers, and maxima/minima represented by the upper/lower lines; '****' indicates $p < 0.0001$.

Figure S4: Ideal observer shows PE bias for both correct and incorrect trials. Ideal observer confidence for correct and incorrect trials, for **(a)** version of PE bias involving manipulation of contrast and noise, and **(b)** version of the PE bias involving superimposed stimuli presented at different contrast levels. All results reflect the probability density over 100 trained networks, with mean confidence in each condition represented by circular markers, and maxima/minima represented by the upper/lower lines; '****' indicates $p < 0.0001$.

Figure S7: Confidence for correct and incorrect trials under alternative training regimes. When trained on the **(a)** standard or **(d)** fixed σ regimes, the model showed a PE bias for both correct and incorrect trials. When trained on the **(c)** fixed μ regime, the model showed a reversed PE bias for both correct and incorrect trials. When trained on the **(b)** fixed μ/σ regime, the model showed a very small PE bias for correct trials, and a very small

reversed PE bias for incorrect trials. These effects canceled each other out, such that there was no overall PE bias for models trained on this regime. All results reflect the probability density over 100 trained networks, with mean confidence in each condition represented by circular markers, and maxima/minima represented by the upper/lower lines; '****' indicates $p < 0.001$, '*****' indicates $p < 0.0001$.

The neural network model should be described in the first subsection of the results section. Currently, the description is entirely in the legend of Figure 1 and methods. A paragraph or two with a slightly extended version of the legend would be helpful.

We have now included a paragraph at the beginning of the results summarizing the model:

Figure 2 illustrates the architecture and training data for our performance-optimized neural network model of decision confidence. The model was trained through standard supervised learning methods both to make a decision about (i.e., classify) an input image, and also to predict the probability that its own decision was correct (Figure 2a). The model was trained on two standard image classification benchmarks, the MNIST and CIFAR-10 datasets, using supervised learning, and was also trained on an orientation discrimination task using reinforcement learning (RL) (Figure 2b). Both contrast and noise level were varied during training, to give the model exposure to a broad range of conditions (Figure 2c). See 'Methods' (Section 4) for more details on the model and training procedures.

The comparison with lesions and stimulations should perhaps be placed nearby in the text to give a more coherent view on the experimental data that could be studied by the ANN approach.

Thank you for this suggestion. We have now combined all of the results pertaining to neural dissociations, including the simulations of the ECoG, TMS, and lesion results, in a single section. Note that we have also included an additional simulation modeling the application of TMS to visual cortex:

2.5.2 Dissociations resulting from brain stimulation

A few studies have found that the application of transcranial magnetic stimulation (TMS) to specific brain regions can have dissociable effects on decisions and confidence. In one study, it was found that low intensity TMS to primary visual cortex (V1) led to a pattern of decreased type-1 sensitivity (d') combined with *increased* confidence [22]. We simulated this experiment in our model by adding random noise (with variance ξ) to the activations in the first layer of the network, which captured the simultaneous decrease in d' and increase in confidence as a result of increasing TMS strength (Figure 9a). Another study found that theta-burst TMS to dlPFC resulted in a greater impairment of type-2 sensitivity (meta- d') than type-1 sensitivity (d') [23]. We simulated this experiment in our model by adding random noise to the activations in the penultimate layer of the network, which captured the greater impact of TMS on meta- d' vs. d' (Figure 9b). These results are both amenable to

standard detection theoretic explanations -- an explanation of the effect of TMS to V1 was proposed in the original study [22], and we present an explanation of the effect of TMS to dIPFC in Supplementary Figure S13. However, they highlight the ability of the neural network model to capture broad anatomical distinctions in the neural mechanisms underlying confidence.

Figure 9: Simulating transcranial magnetic stimulation (TMS) to visual and prefrontal cortices. TMS was simulated by adding noise with variance ξ to activations in a specific layer. **(a)** Simulated TMS to the first layer of the network resulted in both decreased type-1 sensitivity (d') and *increased* confidence, as has been observed following TMS to primary visual cortex (V1). **(b)** Simulated TMS to the penultimate layer of the network resulted in a selective impairment of type-2 sensitivity (meta- d'), as has been observed following TMS to dorsolateral prefrontal cortex (dIPFC). See Supplementary Figure S13 for a detection theoretic explanation of this effect.

[22] D. A. Rahnev, B. Maniscalco, B. Luber, H. Lau, and S. H. Lisanby, “Direct injection of noise to the visual cortex decreases accuracy but increases decision confidence,” *Journal of neurophysiology*, vol. 107, no. 6, pp. 1556–1563, 2012.

The authors do not really distinguish between experiments involving accumulation of evidence and those involving “static” stimuli in their discussions. In accumulation tasks, the amount of “post-decisional” evidence can vary according to experimental design and a direct comparison can be difficult. The authors should highlight those points.

This issue is now highlighted in the final paragraph of the discussion. We hope to investigate this in future work by developing a recurrent version of our model.

For instance, the model could be expanded by replacing the feedforward encoder with a recurrent network [46], allowing the model to make contact with the rich body of data on temporal evidence accumulation in decision-making [47,48], and to account for phenomena such as the effect of post-decisional evidence on confidence judgments [49].

[49] J. Navajas, B. Bahrami, and P. E. Latham, "Post-decisional accounts of biases in confidence," *Current opinion in behavioral sciences*, vol. 11, pp. 55–60, 2016.

Figures 7b, S5c and S7b,c. I would change the color scheme for s1/s2 so it is different than for the panels showing correct/incorrect distributions in the rest of the paper. These panels show different information, and a similar color scheme might confuse the reader.

We have revised all figures (Figures 7, 10, S10, S11, S13, S14) so that they employ a consistent color scheme for correct vs. incorrect trials (green vs. purple) and s1 vs. s2 (blue vs. red).

They should however: 1. Include the full URLs in the methods section rather than links through Hyperlinks. 2. Add some information in the methods about the hardware used for the simulations.

Both of these changes are now reflected in section 4.1:

All code is available at:

https://github.com/taylorwebb/performance_optimized_NN_confidence

All simulations were performed using a single NVIDIA GeForce RTX 2080 Ti GPU.

Reply to reviewer 2

I'm glad of the opportunity to review this paper which I have seen in multiple iterations both as a preprint and conference presentation at CogSci. Each time I see these results I continue to think what an impressive and counterintuitive piece of work it is. I am often skeptical of the value of "black box" machine learning models of cognition, but this one is different in that it targets a common assumption in the field (and one I believe was originally held by the senior author of this paper!) – that a distinct mechanism or representation may need to be posited to explain biases in confidence and metacognition.

Thank you very much for such a positive assessment of our work. You are correct in thinking that, prior to this study, we believed that distinct mechanisms were necessary to account for decisions vs. confidence. We were very surprised by the results!

At a few points in the introduction the RCE rule is introduced as all or none – "...only considers" the response-congruent evidence, or "confidence is based only on the evidence in favour...". This strikes me as an overly strong characterisation of the empirical literature...

We agree that this characterization was too strong. We have now modified these statements:

These dissociations have led to the formulation of an alternative model in which decisions are made according to the BE rule, but confidence is estimated using a simpler heuristic strategy that primarily considers the 'response-congruent-evidence' (RCE; Figure 1b) [11-17]. That is, after weighing the evidence and making a decision, confidence is based mainly on the evidence in favor of the decision that was made.

It struck me that, compared to state-of-the-art image classification, the performance in the MNIST and CIFAR databases was notably low (eg 55% correct in Figure 2d, where chance is presumably 10%). What is the reason for this? Is it artificially constrained by the noise in the image?

This is indeed because of the noise that we added to the images. Without image noise, our model achieves competitive results on both MNIST and CIFAR-10 (~96% on the MNIST test set and ~88% on the CIFAR-10 test set). We specifically targeted (by performing a search over contrast and noise levels) accuracy at the threshold between chance and ceiling performance, which corresponds to 55% accuracy for 10-choice tasks, and 75% accuracy for two-choice tasks. We did this because we expected that this range would be maximally sensitive to any potential differences in confidence, whereas any differences between conditions become compressed as performance gets closer to either chance or ceiling performance. We have now included a note clarifying this issue in section 2.1.1:

Note that stimulus parameters (contrast and noise) were set so as to target the threshold between chance performance and 100% accuracy, resulting in ~55% accuracy for 10-choice tasks and ~75% accuracy for two-choice tasks. The model achieved much higher accuracy when presented with noiseless images (96.3% \pm 0.03 for MNIST, 88.1% \pm 0.05 for CIFAR-10).

"Blindsight patients have very low confidence in their visual discriminations". Is this true? My understanding from the cited Persaud et al. study was that overall confidence (the average proportion of high gambles) was similar in the blind and sighted fields, but that metacognitive sensitivity was impaired.

It is true that, in the study from Persaud et al., patient GY makes a high wager on an approximately equal number of trials for his blind vs. sighted hemifield. However, this most likely reflects a more cognitive level understanding on GY's part that he does in fact have some visual ability in his blind hemifield, and therefore ought to bet on it. More generally, the most commonly used method for assessing blindsight is the 'commentary key' developed by Weiskrantz, according to which blindsight patients routinely describe the visual discriminations made in their blind hemifield as 'guesses' (when given the options 'seen' vs. 'guess'). This is an area that could certainly benefit from more careful study (i.e., to more clearly distinguish between confidence judgments and judgments of 'seeing'), but our view is that the most likely interpretation of the current available evidence is that blindsight patients do indeed have low confidence in their visual discriminations. We have included a footnote addressing this:

It should be noted that, although blindsight patients routinely refer to the discriminations in their blind hemifield as 'guesses' [24], one study found that the blindsight patient GY was willing to place wagers on the visual discriminations made in his blind hemifield at roughly the same rate as those in his intact hemifield [25]. However, this most likely reflects an understanding of his condition at the cognitive level, rather than a genuine sense of confidence. Consistent with this, GY's wagers were not predictive of whether the discriminations in his blind hemifield were correct, as would be expected if they were based on a genuine sense of confidence.

[24] L. Weiskrantz, "Blindsight: A case study and implications," 1986.

[25] N. Persaud et al., "Awareness-related activity in prefrontal and parietal cortices in blindsight reflects more than superior visual performance," *Neuroimage*, vol. 58, no. 2, pp. 605–611, 2011.

Figure 3 – why is meta-d' systematically negative in the lesion condition?

We think that this is due to the highly non-Gaussian nature of the distributions (figure 10b of the revised manuscript). Meta-d' assumes Gaussian distributions, and can display weird behavior when this assumption is violated. As can be seen in figure 10b, the distributions for correct and incorrect trials are almost perfectly overlapping, and so it is very unlikely that confidence has a genuinely antipredictive relationship with correct/incorrect (which is what negative meta-d' would imply). We have added a footnote with this clarification:

It should also be noted that the small negative values of meta-d' in Figure 10a are most likely due to the highly non-Gaussian nature of the confidence distributions, which violates the assumptions of meta-d'.

The analysis in Figure 4 of the drivers of the bias is informative. It struck me that the explanation being put forward is compatible with classical cue-based inferential models of confidence in the memory literature. In other words, the neural network appears to be capitalising on any feature of the input that usually tracks with confidence – similar to how fluency becomes a useful predictor of memory confidence. Is this analogy correct / useful? Ironically, if yes, perhaps the more recent metaperception literature was misled by a focus on Bayesian latent states that are directly coupled to features of the stimulus, rather than cues more broadly construed.

This is a very interesting and important point. These perspectives may not be mutually exclusive. Our proposed explanation for the PE and related biases is that they reflect a strategy that is globally optimal for the broader distribution of sensory experiences (represented by the training data distribution in our model), even if they are not locally optimal for the distribution of sensory inputs in a particular task performed in the laboratory (represented by the test data distribution in our model). Our newly added ideal observer analysis shows that this can be understood from the perspective of Bayesian latent states – the presence of variable contrast in the model's training data has a significant impact on the shape of the sensory distributions, which in turn has a significant impact on the behavior of the ideal observer. However, these

phenomena can also be understood from the perspective of which features in the training data are most diagnostic of accuracy – when contrast varies in the training data, it is a very strong predictor of accuracy. We agree that this seems potentially analogous with the role of fluency in memory confidence, which suggests that it may also be possible to develop an account of the impact of fluency in terms of Bayesian latent states. We have now included the following addressing this connection in the discussion:

There is also an interesting parallel to models of confidence in memory in which fluency serves as a cue to the reliability of a memory [33]. There too, one possible explanation is that this heuristic is employed because fluency is generally a reliable predictor of the accuracy of one's memories [34], even if it is not necessarily diagnostic in every task setting [35].

[33] A. L. Alter and D. M. Oppenheimer, “Uniting the tribes of fluency to form a metacognitive nation,” *Personality and social psychology review*, vol. 13, no. 3, pp. 219–235, 2009.

[34] A. S. Benjamin and R. A. Bjork, “Retrieval fluency as a metacognitive index,” in *Implicit memory and metacognition*, Psychology Press, 2014, pp. 321–350.

[35] C. M. Kelley and D. S. Lindsay, “Remembering mistaken for knowing: Ease of retrieval as a basis for confidence in answers to general knowledge questions,” *Journal of memory and language*, vol. 32, no. 1, pp. 1–24, 1993.

Khalvati et al. (ref. 37) also propose that a common decision variable can underpin dissociations between performance and confidence, including the positive evidence bias. However, they appeal to a somewhat different mechanism – Bayesian inference with partially observable state information. Can this difference in perspective be reconciled or commented on?

We have now included a paragraph in the discussion explicitly addressing the relation between our work and Khalvati et al:

Our model is also related to the recently proposed model of Khalvati et al. [32], in that both models propose a rational account of seemingly suboptimal dissociations between decisions and confidence. An important distinction is that the model of Khalvati et al. depends on a unidimensional representation of sensory evidence, and therefore cannot account for effects that require a two-dimensional sensory evidence space. These include the dissociation between type-1 and type-2 sensitivity, and the version of the PE bias involving superimposed stimuli, both of which our model accounts for. However, a deep commonality is that both models invoke a distinction between the sensory evidence distribution assumed by the experimenter, and the actual distribution used by the decision-maker. In the case of our model, we specifically propose that such dissociations arise because they are globally optimal according to the broader distribution of the decision-maker's prior sensory experiences, even if they are not locally optimal for the sensory evidence distribution in a specific task.

Reviewer comments:

Reviewer #1 (Remarks to the Author: Overall significance):

In this revision, the authors have substantially reframed their manuscript by the addition of an ideal observer model as suggested in the initial review. Overall, the authors have answered most of the points I raised during the initial review.

The revised version of the paper supports the view from other research groups that although there can be an apparent dissociation between confidence reports and choice, this dissociation can be a result of the statistics of the underlying decision variables and actually be a signature of sound statistical computations. By exploring this using an artificial neural network trained not only on choosing the appropriate option but also on estimating the accuracy of decisions, they show that the patterns of confidence reports and metacognitive accuracy (as measured by meta-d') are consistent with an ideal observer model.

This builds on work on decision confidence from several groups (e.g. Hangya et al., *Neural Comp.*, 2016, Fleming and Daw, *Psych Rev*, 2017, Adler and Ma, *Neural Comp.*, 2017, Khalvati et al., *Nature Comms.* 2021) that show the importance of constructing such models of the expected distribution of confidence reports and more generally supports the need to construct such models in cognitive science (Palminteri, Wyart and Koechlin, *Trends in Cognitive Science*, 2017).

Overall, the addition of the Bayesian ideal observer based on a low-dimensional latent representation of the training data provides strong evidence that this process is also occurring in the neural network and in humans performing this task.

Reviewer #1 (Remarks to the Author: Impact):

I think the paper could be suited for publication in *Nature Communications*. Although the idea that choice and confidence use a similar decision variable has been argued for and shown in simpler tasks before, this paper shows it occurs in more complex task. I expect that the approach used here of using a neural network will allow researchers to investigate the expected pattern of confidence reports in complex tasks for which it is difficult to build ideal observer models from normative principles. We can regret the lack of investigation of single neuron representations within the network but the approach proposed here provides a framework for future work in this direction.

Reviewer #1 (Remarks to the Author: Strength of the claims):

Overall, the authors have addressed most of my comments and the revised paper makes a much stronger point through the inclusion of the ideal observer model.

1. The authors have added in Supplementary figures "violin" plots allowing for a better appreciation of the distributions underlying the bar plots in the main paper. Confidence reports are on a graded scale so

the shape of the distribution across conditions is important to understand differences across conditions. I leave it to the editor as to whether these plots should be in the main paper.

2. I would still have liked to see an analysis of some single neuron representations. Presumably neurons in the later layers should have an activation close to direct representation of confidence? But I also understand that this work focuses on the behavioral report of confidence and these representations likely depend on hyperparameter choices of the model.

The author should however extend a bit their discussion of dissociation between choice and confidence across areas. As their model shows, we might not expect dissociations between choice and confidence in single neuron representations, but we might see that these computations are distributed differently even in later layers of the network. They should perhaps also discuss the work from the Miyamoto and Kepecs lab looking at distinct contribution of brain areas to choice and confidence reports despite a common decision variable. For example, the activity of single OFC neurons drives confidence-driven behaviors such as time investments or updating (Masset et al., *Cell*, 2020) and OFC inactivation does not affect decisions but impair confidence-driven time investments (Lak et al., *Neuron*, 2014). Similarly, neural activity in monkey pulvinar reflects the decision variable underlying choice and confidence-driven opt-out decisions but inactivation of pulvinar only impairs opt-out decisions and leaving unimpaired the choice behavior (in contrast to LGN lesions) (Komura et al., 2013, *Nature Neuroscience*). Importantly, this paper suggests the dissociations between representations driving decisions and confidence reports might not be localized only in pre-frontal cortex.

These studies are important as they provide constraints to the in-silico lesion experiments and would guide the study of single “neuron” representations that can be performed by the model network proposed in this paper.

Reviewer #1 (Remarks to the Author: Reproducibility):

1. This paper is based on training a neural network and investigating some of its properties. I commend the authors for providing the full code for their model and simulations on well annotated repository on GitHub. This should enable any researcher with access to a computer with a GPU to reproduce their results and further investigate their model.

Reviewer #2 (Remarks to the Author: Overall significance):

I thank the authors for their comprehensive responses. The addition of the latent ideal observer in response to the important query from Reviewer 1 is very useful, and provides a much clearer picture of why the network is exhibiting these biases.

I have no further concerns. One final suggestion - the authors could consider adding a sentence or two to explain to the reader why the simulation of the Peters et al. data produces the BE-advantage for decisions when decoding from the whole network, but not the penultimate layer. Their current

commentary appeals to the limited spatial resolution of neuroimaging to account for greater contributions of early sensory areas to decision decoding. But this is not an issue when decoding from the network - or do the authors have in mind something about decodability of decision-relevant information differing between early and late layers of the network?

We would like to thank the reviewers for their additional comments and suggestions. We have now revised the manuscript to address these remaining concerns. Specifically, we have added analyses of the model's learned representations at the single unit level, and some additional discussion of remaining issues raised by the reviewers. Below we include a point-by-point reply to the issues raised by the reviewers, along with a description of the corresponding revisions made to the manuscript. The reviewers' comments are presented in blue, and revisions are presented in red.

Reply to reviewer 1

The authors have added in Supplementary figures “violin” plots allowing for a better appreciation of the distributions underlying the bar plots in the main paper. Confidence reports are on a graded scale so the shape of the distribution across conditions is important to understand differences across conditions. I leave it to the editor as to whether these plots should be in the main paper.

We would like to thank the reviewer again for the suggestion to add these plots to the paper, which indeed offer a richer characterization of confidence. We have followed the reviewer's suggestion, and accordingly consulted the editor, who suggested that we keep these new plots in the Supplementary section.

I would still have liked to see an analysis of some single neuron representations. Presumably neurons in the later layers should have an activation close to direct representation of confidence?

We have now included analyses of the representations learned at the single unit level. We found that the penultimate layer of the model contained some neurons that were strongly predictive of decisions, and other neurons that were strongly predictive of confidence. In addition, consistent with previous results, we found that the ‘decision neurons’ implicitly represented confidence, in the sense that they responded more to their preferred stimulus when the network had high confidence. The following addition to the paper describes these results:

Analysis of single unit representations

The representational scheme learned by the model, in which both confidence and decisions are represented using a common decision variable, is reminiscent of findings at the single neuron level in the lateral intraparietal cortex (LIP) [3,5]. In those studies, nonhuman primates were presented with a perceptual decision-making task in which they sometimes had the option to opt out of the decision and receive a small but guaranteed reward (referred to as the ‘sure target’, or T_S), similar to the task that we used when training our model using RL. It was found that LIP neurons were both predictive of decisions, and implicitly encoded confidence, in the sense that they were more active for trials on which the neuron's preferred stimulus (T_m) was chosen vs. trials on which the sure target (T_S) was chosen (Figure 8b). Similarly, these neurons were less

active when their non-preferred stimulus (T_{opp}) was chosen. These neurons thus encoded both decisions and (implicitly) confidence as a single decision variable.

We tested whether our model would show similar effects, by analyzing responses at the single neuron level in the version of the model trained on the RL orientation discrimination task. Specifically, we analyzed the response of individual neurons in the penultimate layer of the network, which showed the same population-level representational signatures as the version of the model trained with supervised learning (i.e., a two-dimensional geometry representing a single decision variable; Supplementary Figure S12). We evaluated the extent to which individual neurons were predictive of either the model's decision output (quantified by $R^2_{decision}$), or the opt-out output (quantified by $R^2_{opt-out}$). We found that some neurons were strongly predictive of decisions (Figure 8a), while other neurons were strongly predictive of the opt-out output (Figure 8c). We classified neurons as either 'decision neurons' or 'confidence neurons', by computing $\Delta R^2 = R^2_{decision} - R^2_{opt-out}$ (see Section 4.6.7 and Supplementary Figure S13). The decision neurons in our model showed a pattern very similar to the behavior of LIP neurons (Figure 8d): they were more active on trials where the preferred stimulus was chosen vs. opt-out trials, and were less active on trials where the non-preferred stimulus was chosen (paired t-tests, $p < 0.05$ for 97 out of 100 trained networks).

It should be noted that, although we treat individual neurons as belonging to discrete categories (decision vs. confidence neurons) in order to compare with previous results, a closer analysis suggests a more distributed pattern. This can be seen by comparing ΔR^2 with the projection of each neuron onto the top 2 PCs (Supplementary Figure S13). Neurons aligned with PC1 were strongly predictive of decisions, while neurons aligned with PC2 were strongly predictive of confidence. But there were also other neurons that interpolated between these clusters, such that they were moderately predictive of both decisions and confidence, and were not strongly aligned with either PC. Thus, although it is possible to view the individual neurons in our model as belonging to discrete categories, a more general interpretation is to view them as jointly representing a low-dimensional geometry at the population level.

Figure 8: Analysis of single unit representations. (a) Example 'decision neuron', strongly predictive of decision output ($R^2_{decision}=0.83$), but not opt-out response ($R^2_{opt-out}=0.004$). (c) Example 'confidence neuron', strongly predictive of opt-out response ($R^2_{opt-out}=0.87$), but not decision output ($R^2_{decision}=0.16$). (b) Kiani & Shadlen (2009) found that decision-making neurons in the lateral intraparietal cortex (LIP) implicitly coded for confidence. They showed higher activity for trials on which their preferred stimulus (T_{in}) was chosen vs. trials on which a 'sure target' (T_S) was chosen (left panel), whereas they were less active for trials on which their non-preferred stimulus (T_{opp}) was chosen (right panel). Each point represents the average normalized (against the pre-stimulus baseline) activation of an individual neuron \pm the standard error of the mean. Histograms depict distribution of distances from diagonal, region shaded in gray represents statistically significant deviations from the diagonal ($p < 0.05$). (d) Decision neurons in our neural network model showed this same pattern. Note that error bars are present on these plots, but are too small to be visible. Neural network results reflect a single example network, but all 100 trained networks displayed a qualitatively similar pattern.

Figure S13: Additional analysis of single unit representations. (a) Confidence vs. decision neurons were classified by computing $\Delta R^2 = R^2_{decision} - R^2_{opt-out}$, with a criterion at $\Delta R^2 = 0$. (b) ΔR^2 was strongly predicted by the projection of each neuron onto the top 2 PCs. Neurons aligned with PC1 ($\Theta \approx 0^\circ$ or $\Theta \approx 180^\circ$) were strongly predictive of decisions ($\Delta R^2 \rightarrow 1$), while neurons aligned with PC2 ($\Theta \approx 90^\circ$ or $\Theta \approx 270^\circ$) were strongly predictive of confidence ($\Delta R^2 \rightarrow -1$). Other neurons were moderately predictive of both decisions and confidence (resulting in lower ΔR^2), and were not strongly aligned with either of the top 2 PCs. Results reflect a single example network, but all 100 trained networks showed a similar pattern.

The author should however extend a bit their discussion of dissociation between choice and confidence across areas. As their model shows, we might not expect dissociations between choice and confidence in single neuron representations, but we might see that these computations are distributed differently even in later layers of the network. They should perhaps also discuss the work from the Miyamoto and Kepecs lab looking at distinct contribution of brain areas to choice and confidence reports despite a common decision variable. For example, the activity of single OFC neurons drives confidence-driven behaviors such as time investments or updating (Masset et al., Cell, 2020) and OFC inactivation does not affect decisions but impair confidence-driven time investments (Lak et al., Neuron, 2014). Similarly, neural activity in monkey pulvinar reflects the decision variable underlying choice and confidence-driven opt-out decisions but inactivation of pulvinar only impairs opt-out decisions and leaving unimpaired the choice behavior (in contrast to LGN lesions) (Komura et al., 2013, Nature Neuroscience). Importantly, this paper suggests the dissociations between representations driving decisions and confidence reports might not be localized only in pre-frontal cortex.

These studies are important as they provide constraints to the in-silico lesion experiments and would guide the study of single “neuron” representations that can be performed by the model network proposed in this paper.

We agree that it is important to more thoroughly discuss how the results of our model at the neural level relate to previous findings. We have now added the following additional discussion:

We found that the representations learned by our model were characterized at the population level by a two-dimensional geometry, in which one dimension coded for decisions, and another

dimension coded for confidence. At the single neuron level, we found that some neurons were both strongly predictive of decisions, and implicitly coded for confidence, in the sense that they responded more to their preferred stimulus class when the network displayed high confidence. These results mirror previous findings from decision-making neurons in LIP [3,5]. In addition to this implicit coding of confidence, we also found that other neurons *explicitly* coded for confidence, in the sense that confidence could be linearly decoded from their activation level, regardless of which choice was made by the network. Such explicit representations of confidence have been discovered at the single neuron level in both orbitofrontal cortex [41] and the pulvinar [28].

A related finding is that a number of brain regions appear to play a selective role in confidence. Temporary inactivation of either orbitofrontal cortex [30] or pulvinar [28] affects confidence-related behaviors without affecting decisions; temporary inactivation of specific prefrontal nodes (areas 9 and 6) has dissociable effects on memory confidence, without affecting memory itself [31]; and lesions to anterior prefrontal cortex cause a domain-specific impairment of perceptual metacognitive accuracy, without impairing perceptual decision accuracy [29]. Our model did not capture this segregation of confidence vs. first-order decision-making capacities into distinct regions. Instead, confidence and decision neurons were intermixed in the penultimate layer of the network, and a simulated lesion to this layer did not cause a selective metacognitive impairment. One likely reason is that, unlike the brain [42,43], there is no pressure in our model for neurons with similar functionality to cluster together spatially. Additionally, our model is missing a number of architectural elements thought to be important for the function of these brain regions, including recurrence, top-down feedback connections, and convergent multimodal inputs. The incorporation of these elements is a promising avenue for future work.

[3] R. Kiani and M. N. Shadlen, “Representation of confidence associated with a decision by neurons in the parietal cortex,” *Science*, vol. 324, no. 5928, pp. 759–764, 2009.

[5] C. R. Fetsch, R. Kiani, W. T. Newsome, and M. N. Shadlen, “Effects of cortical microstimulation on confidence in a perceptual decision,” *Neuron*, vol. 83, no. 4, pp. 797–804, 2014.

[28] Y. Komura, A. Nikkuni, N. Hirashima, T. Uetake, and A. Miyamoto, “Responses of pulvinar neurons reflect a subject’s confidence in visual categorization,” *Nature neuroscience*, vol. 16, no. 6, pp. 749–755, 2013.

[29] S. M. Fleming, J. Ryu, J. G. Golfinos, and K. E. Blackmon, “Domain-specific impairment in metacognitive accuracy following anterior prefrontal lesions,” *Brain*, vol. 137, no. 10, pp. 2811–2822, 2014.

[30] A. Lak, G. M. Costa, E. Romberg, A. A. Koulakov, Z. F. Mainen, and A. Kepecs, “Orbitofrontal cortex is required for optimal waiting based on decision confidence,” *Neuron*, vol. 84, no. 1, pp. 190–201, 2014.

[31] K. Miyamoto et al., “Causal neural network of metamemory for retrospection in primates,” *Science*, vol. 355, no. 6321, pp. 188–193, 2017.

[41] P. Masset, T. Ott, A. Lak, J. Hirokawa, and A. Kepecs, “Behavior-and modality-general representation of confidence in orbitofrontal cortex,” *Cell*, vol. 182, no. 1, pp. 112–126, 2020.

[42] J. Saarinen and T. Kohonen, “Self-organized formation of colour maps in a model cortex,” *Perception*, vol. 14, no. 6, pp. 711–719, 1985.

[43] K. Obermayer, H. Ritter, and K. Schulten, “A principle for the formation of the spatial structure of cortical feature maps.,” *Proceedings of the National Academy of Sciences*, vol. 87, no. 21, pp. 8345–8349, 1990.

Reply to reviewer 2

One final suggestion - the authors could consider adding a sentence or two to explain to the reader why the simulation of the Peters et al. data produces the BE-advantage for decisions when decoding from the whole network, but not the penultimate layer. Their current commentary appeals to the limited spatial resolution of neuroimaging to account for greater contributions of early sensory areas to decision decoding. But this is not an issue when decoding from the network - or do the authors have in mind something about decodability of decision-relevant information differing between early and late layers of the network?

Yes, we agree that some clarification is needed here. In the whole-network decoding analysis, the penultimate layer is included as part of the feature space over which the decoder operates, so in principle the decoder could just ignore the earlier layers and arrive at the exact same result as when decoding from the penultimate layer alone. However, because of the very large feature space involved in decoding from the entire network (11,236 neurons), a decoding method based on gradient descent was necessary, which is susceptible to local minima, and very likely the reason why the decoder converges to a different solution when decoding from the entire network vs. decoding from the penultimate layer only. So, the exact reason why this decoding analysis is sensitive to features in earlier layers is likely not the same reason why real neural decoding analyses are overly sensitive to early sensory features. But the primary point of this analysis is to show that, if a decoding analysis includes a mix of high-level and low-level features, then one cannot necessarily treat this as an evaluation of the high-level decision variable alone. We have included a footnote clarifying why there is a discrepancy between the whole-network and penultimate-layer-only decoding analyses:

In principle, since the penultimate layer forms a part of the feature space in the whole-network decoding analyses, and decisions are nearly perfectly decodable from this layer, the optimal decoder should be able to ignore the earlier layers and produce the same results for both analyses. However, the very large number of features present in the whole-network analysis

(11,236 neurons) necessitated the use of a decoding method based on gradient descent, which is susceptible to local minima, and therefore will not necessarily converge to the optimal result. This likely explains the discrepancy between these analyses.